# Single-cell RNA-sequencing analysis of estrogen- and endocrine-disrupting chemical-induced reorganization of mouse mammary gland

Noriko Kanaya[1], Gregory Chang[1], Xiwei Wu[2], Kohei Saeki[1], Lauren Bernal[1], Hyun-Jeong Shim[1], Jinhui Wang[2], Charles Warden [2], Takuro Yamamoto[1], Jay Li[1], June-Soo Park[3], Timothy Synold[1], Steve Vonderfecht[4], Michele Rakoff[5], Susan L. Neuhausen[6] & Shiuan Chen [1]*

Menopause is a critical window of susceptibility for its sensitivity to endocrine disrupting chemicals due to the decline of endogenous estrogen. Using a surgical menopausal (ovariectomized) mouse model, we assessed how mammary tissue was affected by both 17β-estradiol (E2) and polybrominated diphenyl ethers (PBDEs). As flame retardants in household products, PBDEs are widely detected in human serum. During physiologically-relevant exposure to E2, PBDEs enhanced E2-mediated regrowth of mammary glands with terminal end bud-like structures. Analysis of mammary gland RNA revealed that PBDEs both augmented E2-facilitated gene expression and modulated immune regulation. Through single-cell RNA sequencing (scRNAseq) analysis, E2 was found to induce *Pgr* expression in both *Esr1*+ and *Esr1*− luminal epithelial cells and *Ccl2* expression in *Esr1*+ fibroblasts. PBDEs promote the E2-AREG-EGFR-M2 macrophage pathway. Our findings support that E2 + PBDE increases the risk of developing breast cancer through the expansion of estrogen-responsive luminal epithelial cells and immune modulation.

[1] Department of Cancer Biology, Beckman Research Institute of City of Hope, Duarte, CA, USA. [2] Integrative Genomics Core, Beckman Research Institute of City of Hope, Duarte, CA, USA. [3] Environmental Chemistry Laboratory, Department of Toxic Substances Control, Berkeley, CA, USA. [4] Center for Comparative Medicine, Beckman Research Institute of City of Hope, Duarte, CA, USA. [5] Breast Cancer Care & Research Fund, Los Angeles, CA, USA. [6] Department of Population Sciences, Beckman Research Institute of City of Hope, Duarte, CA, USA. *email: schen@coh.org

During menopausal transition (peri-menopause), endogenous levels of estrogen and progesterone start to decline. Therapy with these hormones eases symptoms of both natural and surgically induced menopause. According to epidemiological studies by the Women's Health Initiative, intervention at this juncture, especially when progesterone is present, may be linked to increased breast cancer incidence[1]. The underlying cellular mechanisms are poorly understood. Estrogen regulates proper development of the mammary gland[2]. Mammary gland cells become hypersensitive to estrogen when its levels are low (after menopause). Thus, this period is considered a critical window of susceptibility; the mammary tissue becomes vulnerable to endocrine-disrupting chemicals (EDCs)[3]. However, little is known about how estrogen replacement and estrogen-like EDCs impact mammary cells, and the relationship between such exposure and the development of breast cancer.

One class of EDCs are polybrominated diphenyl ethers (PBDEs). While PBDE production and use have been curtailed in the US, biomonitoring studies detect them in humans worldwide. PBDEs have been used as flame retardants for plastics, furniture, and electrical equipment[4]. Importantly, these compounds have been associated with various adverse health effects including neurodevelopmental defect, thyroid dysfunction, and reproductive toxicity[5]. By measuring 14 PBDE congeners in the fat tissue of women with and without breast cancer, higher levels of these PBDEs were linked to increased risk of breast cancer[6]. Nevertheless, mechanisms of PBDEs on mammary gland development are inadequately studied.

When mammary ducts grow, estrogen primarily impacts mammary epithelial cells. Mammary ductal expansion involves crosstalk between the epithelium and nearby stromal/immune cells through paracrine signaling[7]. The mammary gland function may be disrupted when unexpected signals are generated. Single-cell RNA sequencing (scRNAseq) has been applied to study mammary gland development[8] and cross-talk mechanisms among luminal, basal, and immune cells[8–11]. These single-cell studies have not yet focused on hormone-regulated changes in mammary glands[12].

Here, we assessed how exogenous estrogen and PBDEs affected the mammary gland in vivo after surgically induced menopause to model hormonal therapies in humans either with or without PBDEs. Our mammary whole mounts show PBDEs augmented estrogen-stimulated regrowth of mammary glands. Using scRNAseq, we identified mammary luminal epithelial cell populations affected by either 17β-estradiol (E2) or a combination of E2 and PBDEs. Our analyses have revealed how these agents act at the single-cell level, mediating phenotypic changes in the mammary gland and affecting luminal epithelial, fibroblasts/stromal, and immune cells. Moreover, they offer fundamental insights into the in vivo activity of PBDEs when estrogen is present. These findings will advance understanding of how exposure to these EDCs impacts the regrowth of mammary glands in the postmenopausal state induced by estrogen.

## Results

**PBDE effects on mouse mammary glands**. Using a surgically induced menopause mouse model, we assessed the impact of exogenous estrogen and/or PBDEs on mammary glands. At 9 weeks old, female BALB/cj mice underwent ovariectomy (OVX, Fig. 1a). The surgery produced expected changes to mammary gland structures, since this model was previously used to study hormone replacement therapy on mammary glands during menopause[13]. OVX mice had mammary glands with regressed branching trees. Four treatments ($n = 10$ in each group) were then initiated: vehicle control, E2, PBDE, and E2 + PBDE. We defined the two groups receiving E2 with or without PBDEs as E2-present groups, and groups without any E2 treatment with or without PBDEs as E2-absent groups. After 1-week treatment, mice were euthanized and mammary glands assessed histologically.

In "E2-absent" groups, whole-mounted mammary glands contained widely scattered ducts, and no terminal end bud (TEB)-like structures (Fig. 1b). These TEB-like structures orchestrate the formation of the ductal tree. (We define these structures as TEB-like rather than TEB because our OVX model was not undergoing through standard mammary gland development.) E2-present groups contained TEB-like structures and larger/thicker ducts; the thicker ducts were often associated with the TEB-like structures, an indication of the enhancement of estrogen response following menopause. For comparison, TEB-like structures were not observed in mammary glands from the Intact mice at the same age (Fig. 1b). Accordingly, E2-present groups exhibited evidence of cellular proliferation. They had higher mitotic figures in both the epithelial cells of mammary ducts and the TEB-like structures. Importantly, mammary fat pads from the E2-present groups exhibited a marker of cellular proliferation (Ki67) in the TEB-like structures, as determined by immunohistochemistry (IHC) analysis (Fig. 1c). While the mammary gland structure from the PBDE-only group did not appear to be different from that of the vehicle control, E2 + PBDE increased the number of TEB-like structures ($n = 10$, Fig. 1c, d). Altogether, the data suggests that E2-induced mammary gland regrowth and PBDEs may modulate the impact of E2 on mammary gland regrowth after menopause. Furthermore, IHC analysis to quantify ducts in mammary glands, through keratin 18 (KRT18) staining, substantiated the observation for the lack of differences between the glands from the vehicle and PBDE-only groups (Fig. 1e, Supplementary Fig. 1). Importantly, the E2-present groups were shown to have more developed (larger) ducts (Fig. 1e).

**Transcriptome analysis using whole mouse mammary glands**. We next assessed potential mechanisms underlying the activity of E2 vs. that of E2 + PBDE using transcriptomic analyses. RNA-sequencing was done on RNA extracted from the fourth mouse mammary gland without the lymph node (vehicle, $n = 3$; PBDE, $n = 3$; E2, $n = 6$; and E2 + PBDE, $n = 6$). Genes differentially expressed between conditions were identified when their fold change > 1.5 and $P$ value ≤ 0.05. Compared to vehicle, mammary glands treated with PBDEs exhibited 0 upregulated genes and 51 down-regulated genes with moderate change of the expression levels. In the E2 group compared to the vehicle group, 558 genes were upregulated and 193 genes were downregulated. Although PBDEs alone had little effect, when PBDEs were co-administered with E2, 608 genes were upregulated and 5 genes were down-regulated compared to E2-only treatment (Supplementary Table 1). The expression of *Pgr* (encodes progesterone receptor, PR) and *Areg* (encodes amphiregulin, AREG), two known estrogen-regulated genes, as well as *Mki67* was upregulated in E2-present groups (Supplementary Fig. 2).

Ingenuity pathway analysis (IPA) network analysis was performed to identify any biological networks affected by E2 and PBDEs. E2 appeared to influence genes in networks associated with cell-to-cell signaling/interaction, cellular assembly/organization and cell cycle, and cellular function/maintenance. In the gene set from the E2 + PBDE group compared with that of vehicle, the above networks were also activated. Furthermore, E2 + PBDE appear to influence genes in networks related to DNA replication. According to these results, mammary epithelial cells from E2 + PBDE may experience an increase in estrogen-mediated cell cycle

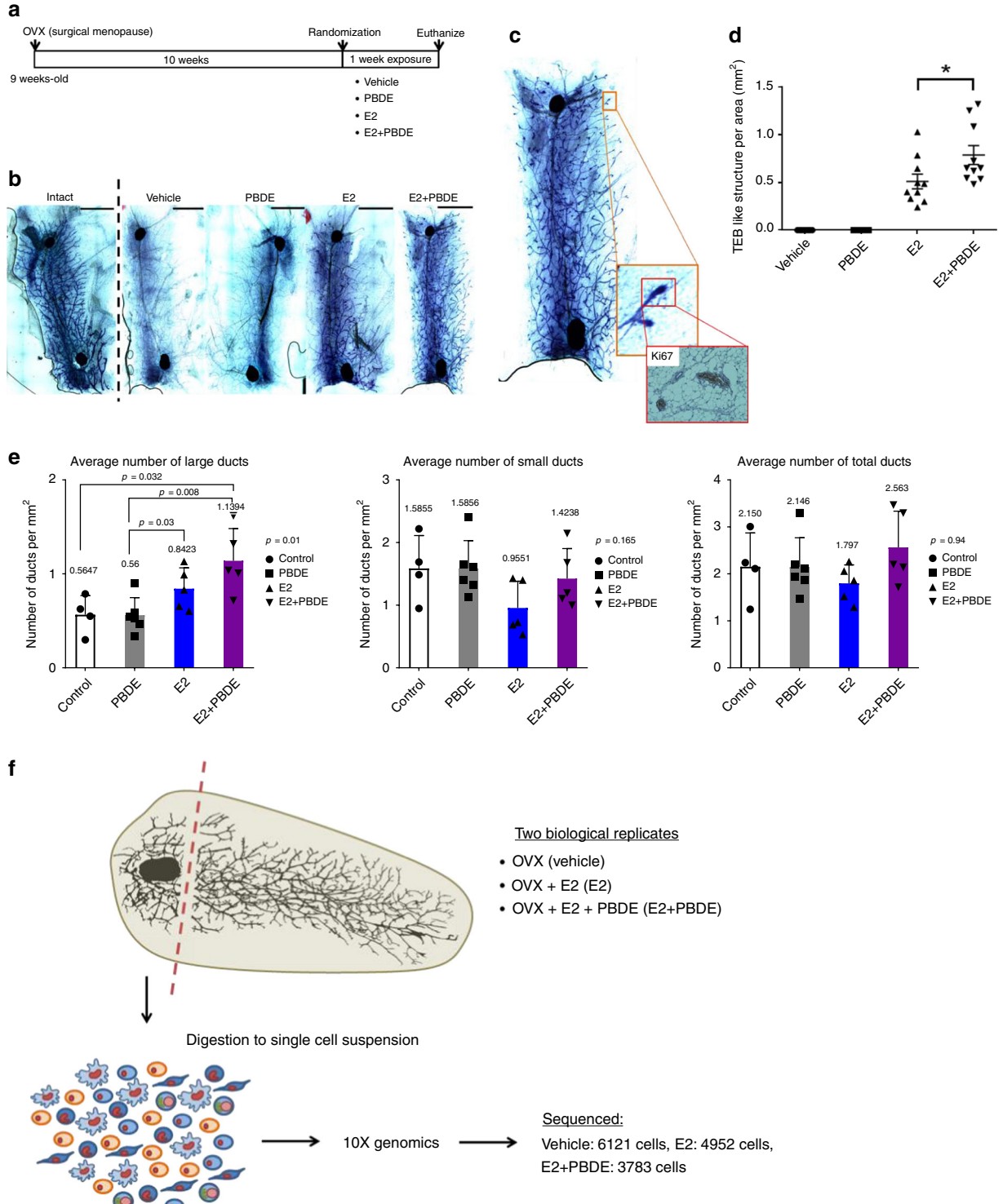

**Fig. 1** Preliminary whole-mount and histological analysis showed differences between vehicle, E2, and E2 + PBDE-treated mice. **a** Overall scheme of experimental design. BALB/cj female mice (9 weeks old) were ovariectomized. Mice were treated ($n = 10$ per treatment) at 19 weeks of age and were euthanized one week after treatment. **b** Toluidine blue stained whole-mounted mammary glands showed that vehicle mice had regressed branching trees as characterized by thin ductal structure by comparing to mammary gland structure of intact mice. Treatment of OVX mice with E2 or E2 + PBDE showed a regrowth of gland structures, characterized by thicker and larger ductal structures and greater prevalence of TEB-like structures. Scale bars = 0.4 cm. **c** TEB-like structures had multiple layers of cells with Ki67-positive staining. **d** The TEB count was significantly increased in mice fed with E2 + PBDE compared with the mice with E2 only. The data are expressed as mean ± SEM of the mean using duplicate assays. Tukey's multiple comparisons test was performed. *$P < 0.05$. **e** Duct quantification by KRT18 immunohistochemistry staining. The number of large, small, and total ducts were counted among vehicle treated ($n = 4$), PBDE treated ($n = 6$), E2 treated ($n = 5$), and E2 + PBDE treated ($n = 5$) mice. A threshold of 1906 μm$^2$ was used to classify ducts as large or small. Ducts that fell below this threshold were classified as small, and ducts that had areas larger than this value were classified as large. **f** Overview of scRNAseq approach using mouse mammary gland samples. Cells from two separate experiments were sequenced on the 10X Genomics platform and then pooled together for analysis

and estrogen-associated signal transduction activities. Additionally, E2 + PBDE had activated genes compared with E2 in the following two networks: hematological system development/function and immune cell trafficking. Thus, PBDEs, in the presence of E2, may be involved in immune cell targeted-effects and epithelial reorganization-effects.

As our results indicated that PBDEs, together with E2, could affect immune modulators, cytokine gene expression was compared among groups using the differentially expressed gene sets. Since mammary gland development involves macrophages, we focused on relationships in the related molecular networks; these include published polarization stimuli and function/secreted molecules[14]. Compared to vehicle, Il10 increased in the E2 group with a fold change of 2.07 ($p = 0.089$); and in the E2 + PBDE group with a fold change of 2.35 ($p = 0.039$). IL10 has been identified as M2 macrophage polarization stimuli[14]. Thus, we hypothesize that the addition of PBDEs to E2 would increase the number of M2 macrophages and/or their activation in mammary glands. The assessment of mRNA expression in whole mammary glands has identified important treatment-associated transcriptional changes. However, the results represent an average across all mammary cell types. They may be dominated by effects in the most abundant type of cells, or overlook effects in smaller cell subpopulations. Thus, further testing of E2-specific and PBDEs-specific effects on mammary gland regrowth would benefit from assessing individual sub-populations of mammary gland cells at the single cell level.

**ScRNAseq to study effects of E2 and PBDEs on mammary glands.** Previous scRNAseq studies on mammary glands focused individually on epithelial or immune cells[8,9]. In this study, we have tried to assess both cell populations along with surrounding stromal cells altogether. The PBDE-only samples were omitted because of their mammary gland phenotype and RNAseq results were similar to the vehicle samples (Fig. 1b–e, Supplementary Table 1). Following treatment and euthanasia, mammary glands were surgically removed and digested into a single-cell suspension. Single-cell samples were prepared for scRNAseq using a 10x Genomics platform. For our study, duplicate single-cell preparations from two independent experiments were processed for scRNAseq. For the three treatment groups, 2291 genes were detected on average from 14,853 cells: 6121 cells of vehicle, 4952 of E2, and 3783 of E2 + PBDE treatments (Fig. 1f). After 1318 highly variable genes (HVGs) were identified, principal component analysis was performed and the top 11 principal components were used for clustering. A t-distributed stochastic neighbor embedding (t-SNE) plot was used to visualize the data in a two-dimensional subspace, which led to identification of 11 major clusters (C0–C10) (Fig. 2a). The number of cells in each cluster are in Supplementary Table 2.

**Changes of the mammary gland at single cell resolution.** To characterize these 11 clusters (Fig. 2a), the relationships between clusters were visualized by a heatmap using the top 5 differentially expressed genes (DEGs) and hierarchical clustering based on average expression of HVGs (Fig. 2b, c).

Annotation of cell types for each cluster was accomplished by comparing the DEGs to previously reported cellular markers (Fig. 2d, Supplementary Fig. 3, Supplementary Data 1). For example, Pal et al.[8] used scRNAseq to construct a developmental lineage for mouse mammary glands. Their list of markers (progenitor, intermediate, and mature luminal) was used to define our cell populations. Likewise, we considered results from Bach et al., who used scRNAseq to determine the gene expression profile of mammary epithelial cells at four developmental stages;

similarly, Nguyen et al. used scRNAseq to reveal three distinct populations (one basal and two luminal) of human breast epithelial cells[9,10]. Expression of Esr1 that encodes estrogen receptor α, a definitive target of E2, was visualized as well (Fig. 2d, Supplementary Fig. 3). Our putative cluster identification was summarized in Table 1 and briefly discussed below.

*C4, C5 (luminal cells):* C4 and C5 are luminal epithelial cells, based on their expression of Krt18, Krt19, and Epcam (Fig. 2d, Supplementary Fig. 4). C4 expressed mature luminal markers[8,10] Ptn, Ly6d, Cited1, Wafdc2, Prlr, Areg, Krt8, Krt7, Sct2, Aqp5, Cd164 (Table 1, Supplementary Fig. 4). C5 expressed more progenitor marker genes[8], such as Csn3, Plet1, Trf, Lcn2, Kit[8,9] (Table 1, Supplementary Fig. 4). Both C4 and C5 clearly express Esr1. To further define them, cells in these two clusters were re-clustered.

*C0–C3 (ECM/fibroblasts):* These cells express genes encoding for collagen and extracellular matrix (ECM)-related molecules, such as Col1a1, Col1a2, Col6a1, Mmp2, and Tnxb, which is a characteristic gene signature of stromal fibroblasts (Fig. 2d, Supplementary Fig. 5). Importantly, Esr1 was also expressed in these clusters (Fig. 2d, Supplementary Fig. 3). Cells were re-clustered to further examine fibroblast/stromal cell variability.

*C6, C8, and C9 (Immune cells):* Ptprc encodes for the leukocyte-associated protein tyrosine phosphatase receptor C, or CD45 (Fig. 2d) which is commonly used for leukocyte identification[15]. Due to the diverse variety of immune cells, these cells were also re-clustered.

*C7 (Basal cells) and C10 (Endothelial cells):* C7 expressed both the basal marker Krt14 and myoepithelial cell marker Acta2. Additionally, C7 expressed smooth muscle cell markers Krt5, Mylk, Myl9, and Tagln[10] (Supplementary Fig. 6). Thus, we concluded that C7 contained basal epithelial cells. C10 expressed genes associated with the endothelial cell compartment: Id1[16], Pecam1, Egfl7[17], Apold1[18], Plvap[19], Lrg1[20], Flt1, and Fabp4 (Supplementary Fig. 6) and thus labeled as endothelial cells (Table 1).

**The impact of E2 and PBDE treatments on C0–C10 clusters.** We assessed the impact of vehicle, E2, or E2 + PBDE on cluster distribution (Fig. 2e, f, Table 1). As summarized in Fig. 2f, Supplementary Fig. 7, the differences from two independent experiments did not affect the assignment of 11 clusters after effective removal of batch effects by canonical correlation analysis (CCA) and subspace alignment.

According to our results, most vehicle-treated cells were in fibroblast clusters; the reduced epithelial cell populations may be due to long-term hormone deprivation. In vehicle-treated mice, mammary glands contained few, widely scattered ducts (Supplementary Fig. 1) and lacked TEB-like structures; most of the area was occupied by stroma-like structures. E2 + PBDE-treated mice had significantly more large ducts ($p = 0.032$) (Fig. 1e). Meanwhile, the numbers of small and total ducts were not significantly different by treatments.

Compared to vehicle, E2 treated resulted in more cells in a unique fibroblast cluster (C2), luminal cell clusters (C4 and C5), and basal cell cluster (C7). This is consistent with previous findings that mammary glands from OVX mice had low expression of Epcam that only increased after E2 treatment[21] (Supplementary Fig. 1). Compared to E2-only treatment, E2 + PBDE showed a trend of cell population increase in luminal clusters C4 and C5, along with a slight increase in basal cells (C7). It was also observed that treatment impacted on immune cell clusters (C6) (Fig. 2f).

Overall, E2 as well as E2 + PBDE increased luminal cells, and changed fibroblast and immune cell clusters. In the sections below, we re-clustered and discussed these populations in more detail.

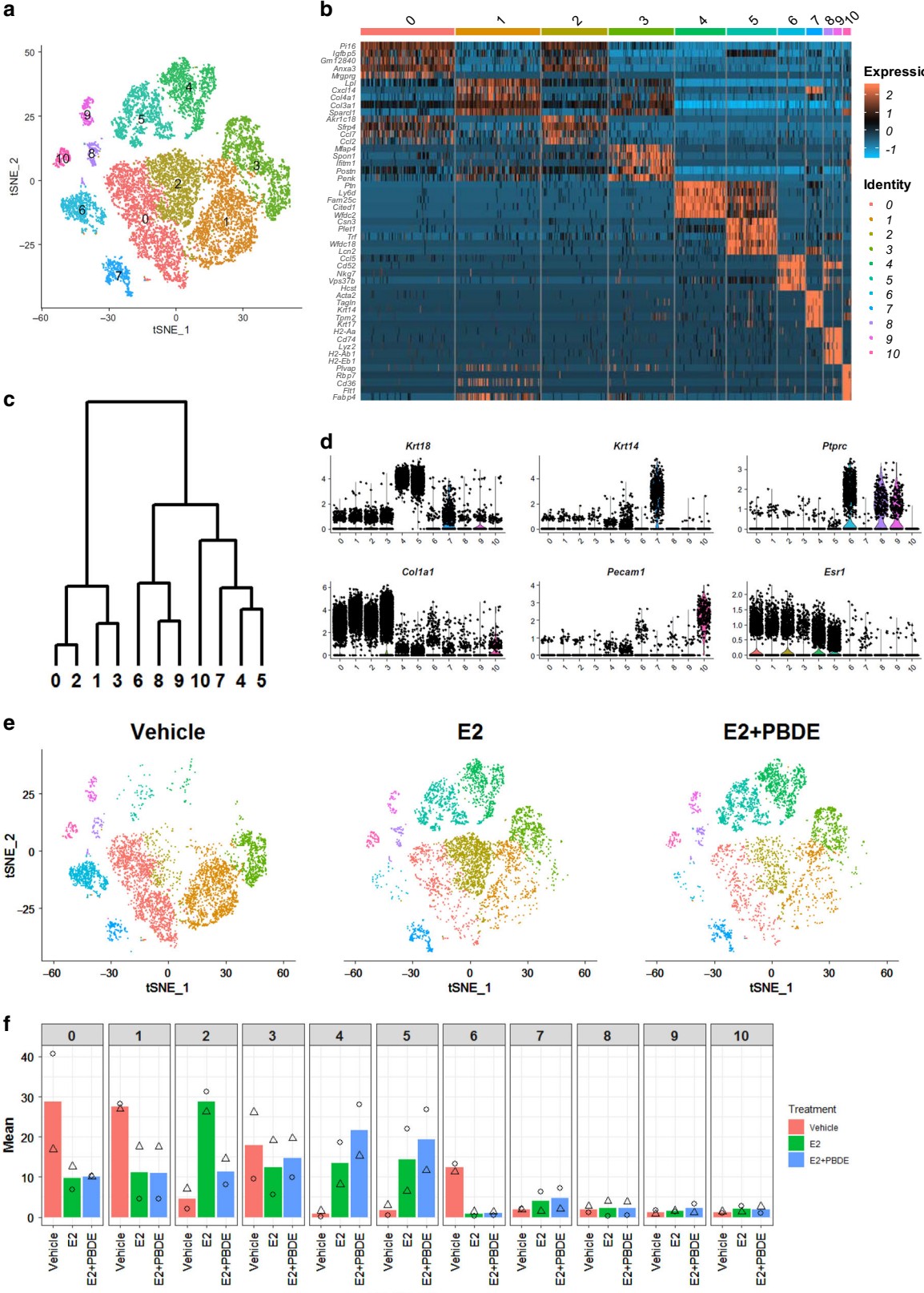

**Fig. 2** ScRNAseq yielded 11 different clusters representing different populations of luminal, fibroblast, immune, and other cells. **a** T-SNE plot of cells from the combination of two independent experiment sets from vehicle, E2 and E2 + PBDE group. Cells were color coded according to clusters. **b** Heatmap showing specific genes (top 5 differentially expressed genes in each cluster, top 20 gene lists shown in Supplementary Data 1). **c** Dendogram showing cluster relationship to other clusters based on gene expression data. **d** Violin plots of gene expression of selected genes unique for each cluster. *Krt18* (luminal cell marker), *Krt14* (basal cell marker), *Ptprc* (immune cell marker), *Col1a1* (fibroblast marker), *Pecam1* (endothelial cell marker), were used to identify distinct cell populations. **e** T-SNE plots of cells separated by treatment group. **f** Cell distributions from two independent experiments by percentage for all three treatment groups among all 11 total clusters. Symbols indicate individual cell distributions for each of the two experimental replicates (triangle, circle)

**Table 1 Summary of clusters and expressed genes**

| ID | Vehicle[a] | E2[a] | E2 + PBDE[a] | Key genes | Cell type |
|----|---------|------|-----------|-----------|-----------|
| C0 | 33.4 | 9.8 | 10.1 | Col1a1, Col1a2, Col6a1, Mmp2, Tnxb | ECM/fibroblast |
| C1 | 27.8 | 11.2 | 9.9 | Col1a1, Col1a2, Col6a1, Mmp2, Tnxb | ECM/fibroblast |
| C2 | 3.5 | 28.7 | 10.8 | Col1a1, Col1a2, Col6a1, Mmp2, Tnxb | ECM/fibroblast |
| C3 | 14.6 | 12.5 | 13.9 | Col1a1, Col1a2, Col6a1, Mmp2, Tnxb | ECM/fibroblast |
| C4 | 0.6 | 13.3 | 22.7 | Krt18, Krt19, Epcam, Cited1, Areg | Mature luminal cells |
| C5 | 1.2 | 14.1 | 20.6 | Krt18, Krt19, Epcam, Csn3, Kit | Luminal progenitor/mature luminal cells |
| C6 | 12.7 | 0.8 | 0.9 | Ptprc | Immune cells |
| C7 | 2.0 | 3.9 | 5.1 | Acta2, Krt14, Krt5, Mylk, Myl9, Tagln | Basal cells |
| C8 | 1.6 | 2.2 | 1.9 | Ptprc | Immune cells |
| C9 | 1.5 | 1.5 | 2.4 | Ptprc | Immune cells |
| C10 | 1.1 | 2.0 | 1.7 | Id1, Pecam1, Egfl7, Apold1, Plvap, Lrg1, Fabp4 | Endothelial cells |

[a]Values are presented as percentages of total number of cells per treatment group (e.g., 33.4% of cells from vehicle-treated group are clustered in C0)

**Table 2 Hormone receptor status of luminal cells**

|    | ER+/PR− | ER−/PR+ | DN | DP |
|----|---------|---------|----|----|
| L0 | 215[a] | 445 | 249 | 499 |
| L1 | 123 | 240 | 94 | 670 |
| L2 | 33 | 20 | 381 | 13 |
| L3 | 18 | 30 | 14 | 59 |

[a]Values represent number of cells

**Re-clustering of luminal epithelial cells**. For better evaluation of luminal epithelial cells (C4 and C5) (Fig. 3a), additional clustering was performed, resulting in four independent clusters (L0–L3) (Fig. 3b). A heatmap (Fig. 3c) and a dendrogram (Fig. 3d) were generated to compare relationship and gene expression profiles among four luminal cell clusters. Expression of known markers and hormone receptors were examined (Fig. 3e, Supplementary Data 2).

L2 was considered as luminal progenitor cells (Kit, Elf5, and Cd44), while L0 and L3 were mature luminal cells (Figs. 3c–e) characterized by Prlr and Cited1. L1 cells expressed genes associated with both mature and progenitor cells, and thus defined as intermediate luminal cells. L2 (luminal progenitor cells) had very few Esr1+ and/or Pgr+ cells, while L0 and L3 (mature luminal cells) had the most Esr1+ and/or Pgr+ cells (Fig. 3e, Table 2).

We also performed Molecular Signatures Database (MSigDB) analysis at the single cell level, which uses collections of annotated gene sets to estimate gene expression profile scores for each cell[22]. Among the four luminal cell clusters, L1 cells had more activated pathways in a subset of the cancer hallmarks functional genes (Fig. 3f).

Changes in cluster distribution by treatments are summarized in Fig. 3g, h, Supplementary Fig. 9a. E2 treatment dramatically expanded luminal cells compared to the vehicle cells. E2 + PBDE did not result in any new sub-clusters. Furthermore, due to regression of mammary glands, the vehicle group had fewer luminal cells, but the distribution of the cells in different luminal clusters was similar to E2 or E2 + PBDE treatments (Supplementary Fig. 10a).

**Esr1+/Pgr+ and Pgr+ cells increased with E2+PBDE treatment**. Since both ERα and PR are critical in the underlying activities of exogenous E2 and PBDEs on mammary gland growth, we further examined the impact of the treatments on Pgr and Esr1 expression (Figs 4a, b). Esr1 expression values were

slightly reduced by E2 and E2 + PBDE treatments (Fig. 4a). Conversely, E2 treatment increased Pgr+ cells, mainly in L0 and L1 (Fig. 4b); Pgr was expressed in both Esr1+ and Esr1− cells (Fig. 4c). When E2 was supplemented with PBDEs, the number of Pgr+ cells further increased (Fig. 4b; Supplementary Table 3).

We examined the histological distribution of hormone receptor-positive cells in mammary glands using immunofluorescence (IF) staining. In vehicle-treated glands, fewer developed ducts were observed; ductal cells were ERα+ (Fig. 4d). As indicated by DAPI staining, ERα was predominantly localized in the cytoplasm (Fig. 4d, vehicle). We also observed stromal cell area to be ERα+. For E2 treatment, ducts exhibited the presence of PR+ cells (Fig. 4d, green) and cells co-expressing ERα and PR (Fig. 4d, yellow). Most glands from the E2-present groups had cells with ERα and/or PR localized in the nucleus of luminal cells; many of these cells were found in TEB-like structures (Fig. 4d). Both TEB-like structures and ducts from the E2-present groups expressed Ki67, as seen by IHC (Fig. 4e, f).

**Gene expression profiles of Esr1+ and Pgr+ cells**. According to the feature plots of Esr1 and Pgr in luminal epithelial cells (Fig. 4a, b) and Table 2, L0/L3 and L1 exhibited cells with four different types of expression patterns: double positive (DP) = Esr1+Pgr+; Esr1 only = Esr1+Pgr−; Pgr only = Esr1−Pgr+; and DN = Esr1−Pgr−. E2 or E2 + PBDE treatment increased Esr1−Pgr+ as well as DP cells.

The mammary gland of the vehicle group contained all four cell types (Supplementary Table 3), but with fewer cells, consistent with a less defined ductal structure (Supplementary Fig. 1). E2 treatment, as it promoted the regrowth of mammary gland with TEB-like structures, led to more luminal epithelial cells overall. E2-induced Pgr expression in both Esr1− and Esr1+ cells, resulting in an increase in Pgr-only and DP cells in the E2-treatment group. There were more Pgr-only and DP cells from E2 + PBDE treatment compared to the E2 treatment group, suggesting that PBDEs enhance the E2-mediated effect (Fig. 4c, Supplementary Table 3).

As AREG has been suggested to mediate estrogen-induced and progesterone-induced development of mammary ducts[23], we assessed how E2-present treatments affected the distribution of Areg+ cells. Areg expression was mostly observed in L0/L3 and L1 (Fig. 4g), confirming that ERα expressed in luminal cells was functionally active. With E2 treatment, Areg+ cells dramatically increased (Fig. 4h). Addition of PBDEs to E2 further increased Areg+ cells (Fig. 4h). Most Areg+ cells showed Pgr expression (Supplementary Fig. 11). Therefore, we hypothesize that PBDEs

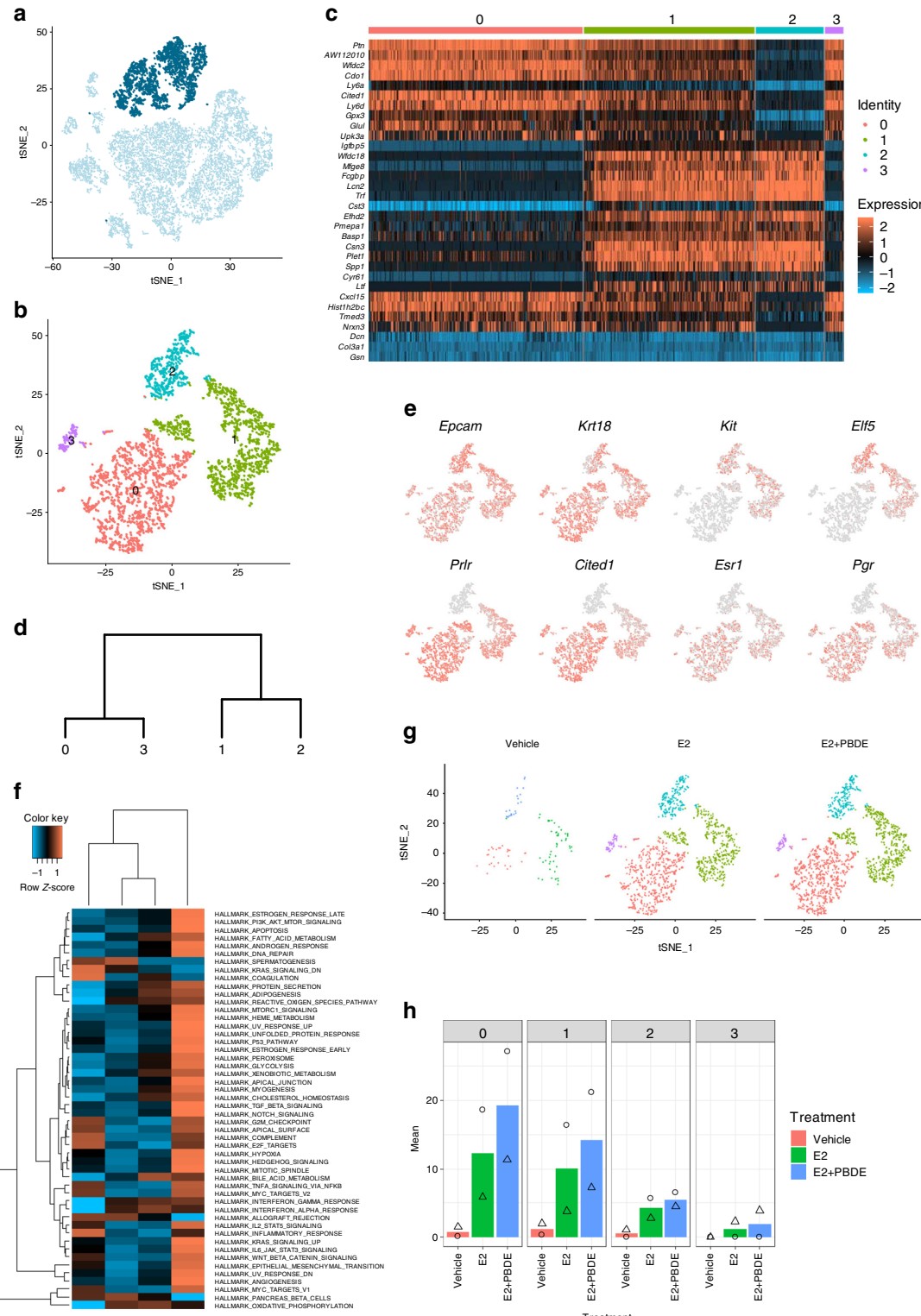

**Fig. 3** Luminal cells were re-clustered to reveal three subpopulations. **a** Feature plot highlighting clusters C4 and C5, which were putatively identified as luminal epithelial cells based on established epithelial markers, such as *Krt18* and *Epcam*. **b** T-SNE plot of re-clustered luminal cells shows four distinct clusters. **c** Heatmap showing specific genes (top 5 differentially expressed genes in each luminal cluster). **d** Dendrogram showing luminal cluster relationship to other luminal clusters based on gene expression data. **e** Selected feature plots of genes used to classify the subpopulations of luminal cells into luminal progenitor cells, intermediate luminal cells, and mature luminal cells. **f** Heatmap on the MSigDB analysis based on the average normalized expression values of genes in the four luminal cell clusters. L1 had the most upregulated pathways relative to the other luminal clusters. **g** T-SNE plots of luminal cell clustering separated by treatment group. **h** Cell distributions from two independent experiments by percentage for all three treatment groups among all four luminal cell clusters. Symbols indicate individual cell distributions for each of the two experimental replicates (triangle, circle)

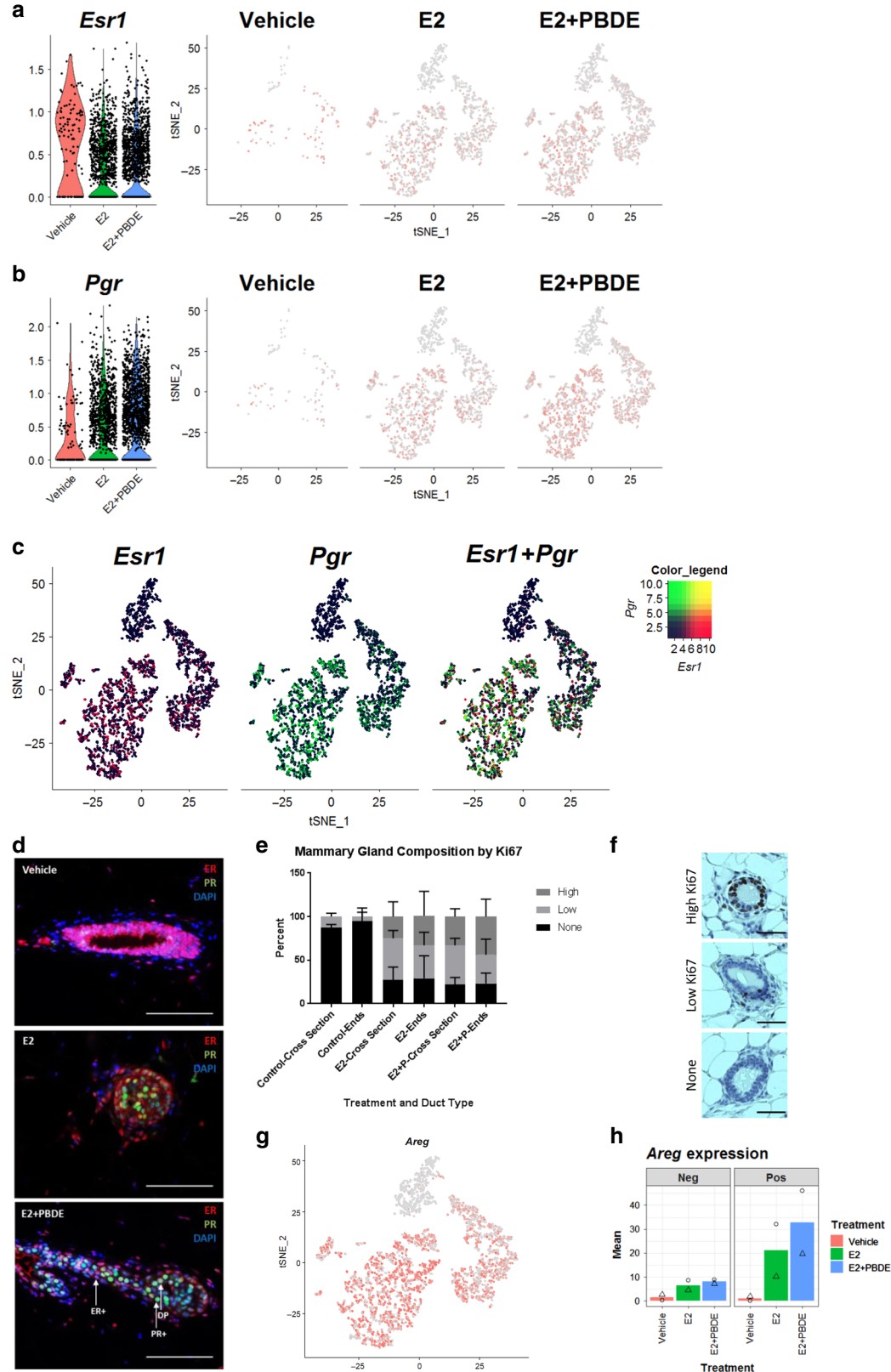

enhance E2 effects by increasing the expression of important master regulators for mammary gland development, e.g. *Areg* and *Pgr*. The results also indicate that estrogen-regulated genes can be expressed in *Esr1*⁻ cells, which implicates paracrine regulatory mechanisms.

**Re-clustering of ECM cells/fibroblasts**. To further characterize ECM/fibroblasts, additional re-clustering was performed. Fibroblasts were negatively selected as done in Bartoschek et al., by selecting cells not expressing *Epcam, Cd45, Cd31, Ng2, Cd52,* and *Krt14* (Fig. 5a)[24]. After re-clustering, we identified seven

**Fig. 4** PBDEs enhance E2 effects by acting on key regulators of mammary gland development. **a** Violin plot and feature plots of *Esr1* expression in the luminal cell clusters. Vehicle-treated cells showed higher expression of *Esr1* relative to E2 and E2 + PBDE-treated cells. **b** Violin plot and feature plots of *Pgr* expression in the luminal cell clusters. *Pgr*+ cells and *Pgr* expression increased in E2-present cells relative to vehicle-treated cells. **c** Feature plots showing co-expression of *Esr1* and *Pgr*. The color code represents the binned and scaled expression data. *Pgr* expression is not specific to only *Esr1*+ cells. **d** Immunofluorescence analysis of ERα (red), PR (green), and DNA stain using DAPI (blue). Arrow indicates each cell types; ER+ only (red), PR+ only (green), and DP (yellow). Scale bars = 100 μm. **e** Quantification of Ki67+ ducts on whole mammary gland sections. IHC slides of mammary glands from vehicle (*n* = 4), E2 (*n* = 4), and E2 + PBDE (*n* = 6) were stained for Ki67. Transverse represents ducts that have a clear lumina, and end buds were defined as ducts that have no clear lumina and are densely packed. High Ki67 levels were defined as ducts that have most of its cells positively stained for Ki67, while low Ki67 levels were defined as ducts that are primarily Ki67−, but do have a few stained cells. **f** Representative images of ducts classified as high Ki67, low Ki67, and none for quantification. Scale bars = 50 μm. **g** Feature plot of *Areg* expression in luminal cell clusters. *Areg* expression was mostly specific to the intermediate and mature luminal clusters (L1, L2). **h** Bar plots of results from two independent experiments for luminal cells expressing *Areg*. PBDE treatment in addition to E2 further increased the number of *Areg*+ cells. Symbols indicate individual cell distributions for each of the two experimental replicates (triangle, circle)

fibroblast/ECM clusters (F0–F6) (Fig. 5b). The distribution of cells per treatment group was shown in Fig. 5c, Supplementary Fig. 8b. A dendrogram (Fig. 5d) and a heatmap (Fig. 5e) were generated to compare gene expression profiles among seven fibroblast/ECM clusters. DEG and MSigDB analyses were performed (Fig. 5f, Supplementary Data 3). F0 and F2 had more *Esr1*+ cells and potentially activated more pathways than other fibroblast/ECM clusters (Fig. 5f, g). These two clusters, especially F2 (from E2-treatment group), had more *Ccl2*+ cells (Fig. 5g). F5 and F6 were mainly from the vehicle group and relatively inactive in terms of GSVA scores (Fig. 5f).

E2 treatment resulted in a unique cluster (C2) that contained *Esr1*+ ECM/fibroblasts. The expression of *Esr1* in mammary stromal cells have been reported[13,25] and suggested to be one of the essential factors for E2-induced luminal epithelial proliferation[13]. As shown in Fig. 5g, the *Esr1*+ fibroblasts had higher expression of *Ccl2*, which was associated with M2 macrophage recruitment and polarization[26], that in turn plays a role in mammary gland development and cancer promotion[27–30].

*Egfr* expression in the ECM/fibroblasts (Supplementary Fig. 12) suggested cross-talk between *Areg*+ luminal epithelial cells and *Egfr*+ fibroblasts. AREG-EGFR-signaling was reported to be important for E2-induced recruitment of macrophages and eosinophils, regulating ductal outgrowth[31].

**Re-clustering of immune cells**. To further characterize immune cells, C6, C8 and C9 were re-clustered to 11 clusters (I0–I10) (Fig. 6a, b). Heatmap of top 20 DEGs and hierarchical clustering were used to visualize the relationships between clusters (Fig. 6c, d). Identities of immune cells were assigned based on published information on the top 20 DEGs (Fig. 6c, Table 3, Supplementary Data 4) and known immune cell markers (Fig. 6e, Table 3).

Three major types of immune cells are discussed below. The first group is T cell and natural killer (NK) cell clusters (I0, I1, I3, I7, and I9): They were defined based on expression of T cell marker *Cd3e*. A portion of I0 lacked expression of *Cd3e*, but expressed well-known NK cell marker *Klrb1c*, or *Nk1.1*, in a mutually exclusive manner. A large portion of I3 also lacked *Cd3e* expression; however, expression of *Gata3*, a master transcriptional factor for T cell differentiation, was observed. As a result, I0 was labeled as T cells/NK cells, and the other four clusters were considered as T cells with different function or differentiation status. The second group is antigen-presenting cell (APC) clusters (I2, I4, I5, and I6): They were designated by expression of *H2-Ab1*, a murine MHC class II molecule. In these four clusters, dendritic cell (DC) marker *Cd24a* was found in I2 and I4. B cell marker *Cd19* was expressed in a part of I4. I5 and I6 were positive for macrophage marker *Fcgr1*, or *Cd64*. To further distinguish differences between macrophage clusters I5 and I6, another DEG analysis was carried out specifically between these two clusters

which identified *C1q* genes as DEGs in I6 (Fig. 6f). *C1q* expression in macrophages has been associated with phenotypic polarization towards M2 macrophages. Overall, I2, I4, I5, and I6 were identified as DC, DC/B cells, macrophages, and M2 macrophages, respectively. The third group is fibrocyte clusters (I8 and I10): The expression of *Cd34*, *Cd45*, and *Col1a1* was found in these cells. The function of fibrocytes have not totally defined, but they have been reported as a novel myeloid-derived suppressor cell subset[32].

E2 and E2 + PBDE treatments increased I6 cells, which was identified as an M2 macrophage cluster (Fig. 6g, Supplementary Figs. 9c and 10c), suggesting a possible link between M2 macrophage increase and an *Il10* expression level increase observed in our whole gland RNAseq. Comparing *Il10* and GSVA scores for BIOCARTA_IL10 between macrophage clusters I5 and I6, *Il10* expression was only found in I6, and I6 had a higher GSVA score (Supplementary Fig. 13). These support our hypothesis that E2 and PBDE treatment causes polarization toward M2 macrophages.

## Discussion

Women are often prescribed estrogen supplements to ease symptoms of natural menopausal transition and surgically induced menopause. However, when estrogen levels decline, mammary glands become hypersensitive towards estrogen. At this juncture, cells become vulnerable to estrogen supplements and EDCs. Brown et al. have reported that obese postmenopausal women could have higher incidence of hormone-dependent postmenopausal breast cancer[33], due to elevated aromatase expression/estrogen production in mammary tissue by adipose dysfunction, supporting cancer promoting effect of non-physiological levels of estrogen after menopause. To define the risk associated with exposure to estrogen and PBDEs during the menopausal transition, it is critical to understand how they affect mammary gland cells.

After OVX of female BALB/cj mice, we found few rudimentary epithelial trees with thin ducts with fewer luminal epithelial cells. We counted the number of TEB-like structures after treatment with either vehicle, E2, a mixture of PBDEs (BDE-47, BDE-100, and BDE-153), or a combination of E2 and PBDEs. TEBs have been used as a model for cancer; they are highly proliferative structures with heterogeneous cell populations and invasive characteristics[34]. As such, TEBs are typically origin points of mammary cancers in both rodents and humans. Moreover, TEBs are sensitive to chemical carcinogenesis, and a higher level of TEBs was associated with an increased risk of developing breast cancer[35]. Here, using the number of TEB-like structures as a primary endpoint, compared to vehicle, E2 increased the number of TEB-like structures. Importantly, the impact of E2 was enhanced with PBDEs, which saw an increase of developed ducts

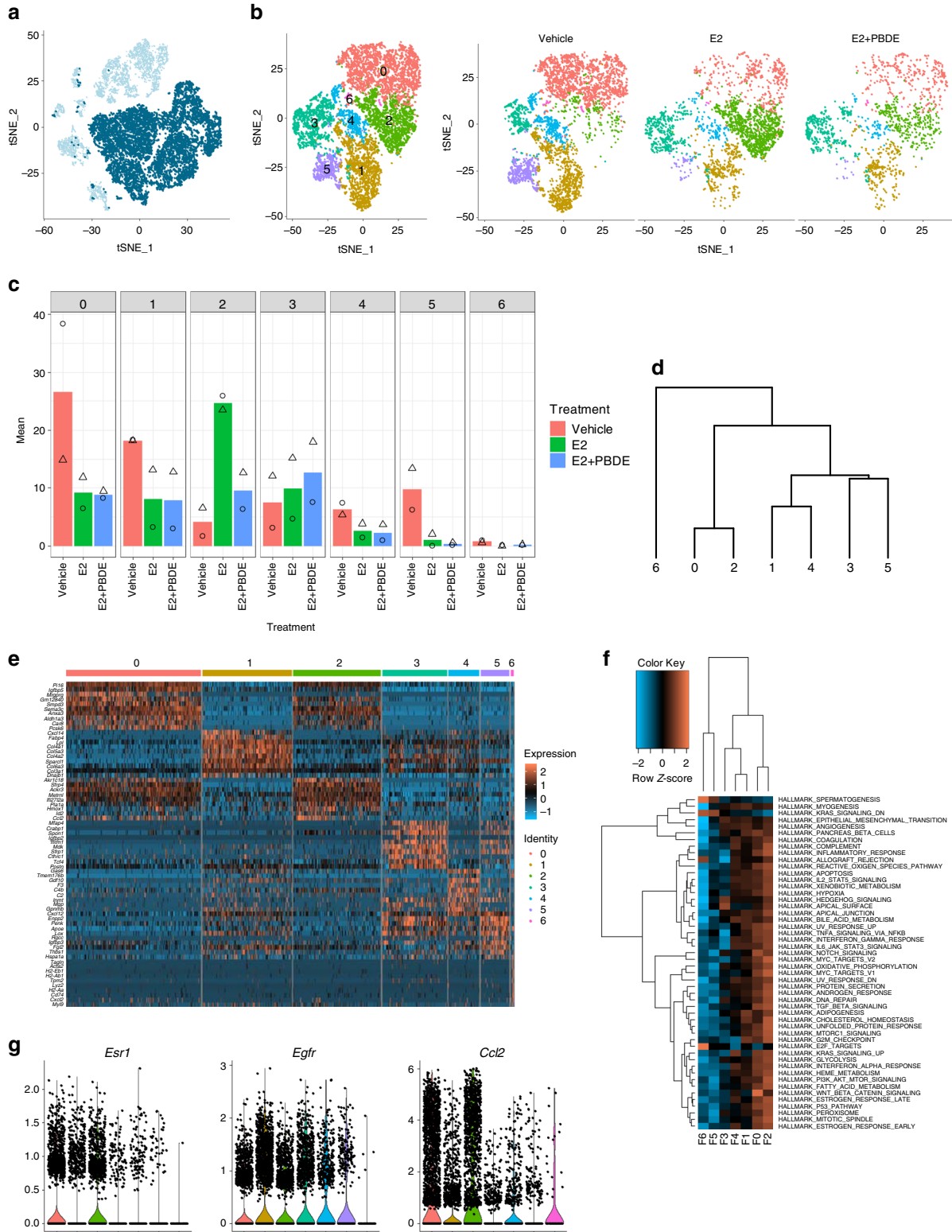

**Fig. 5** ECM associated and fibroblast cells were re-clustered to show a specific subpopulation of *Ccl2*+ cells. **a** Feature plot highlighting *Epcam*, *Cd45, Cd31, Ng2, Cd52*, and *Krt14* negative cells to negatively select for ECM and fibroblast cells. **b** T-SNE plots of re-clustered ECM/Fibroblast clusters. **c** Bar plots showing the distribution of cells from two independent experiments in each ECM/Fibroblast cluster. Symbols indicate individual cell distributions for each of the two experimental replicates (triangle, circle). **d** Dendrogram showing the relationship of each ECM/fibroblast cluster with other clusters. **e** Heatmap showing the top 5 differentially expressed genes for each of the 7 ECM/Fibroblast clusters. **f** Heatmap on the MSigDB analysis based on the average normalized expression values of genes in the 7 ECM/Fibroblast cell clusters. Cluster F2 had the most upregulated pathways relative to all other fibroblast clusters. **g** Violin plots of selected genes emphasizing the unique gene signature of *Esr1*+ ECM/Fibroblast cells. Clusters F0, F1, and F2 had more cells with higher expression of *Esr1* and *Ccl2*, whereas *Egfr* was observed in nearly all fibroblast clusters

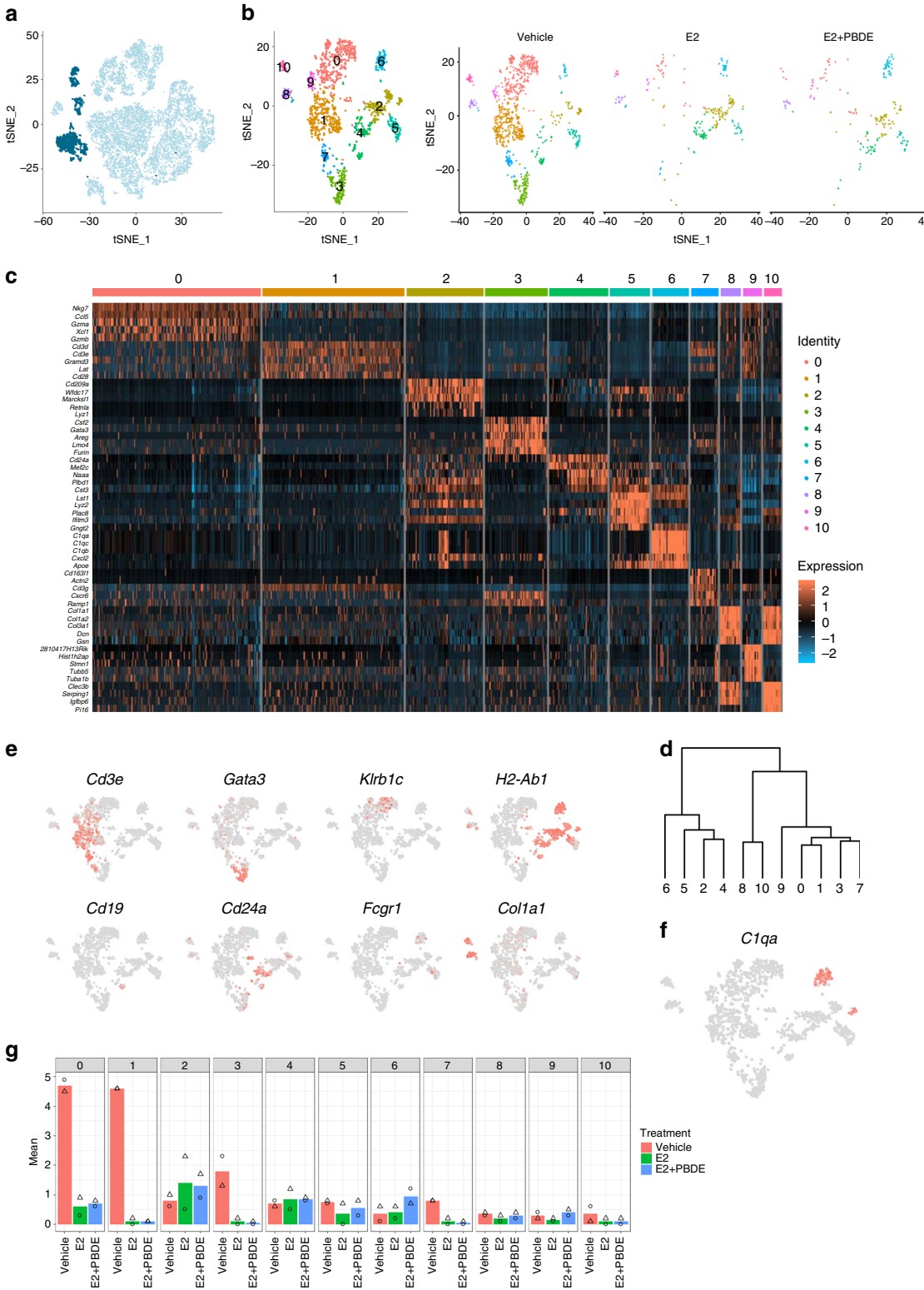

**Fig. 6** Re-clustered immune cell clusters C6, C8, and C9 resulted in 11 subclusters. **a** Feature plot highlighting C6, C8, and C9, which were putatively identified as immune cells based on high *Ptprc* expression in these clusters. **b** T-SNE plots of re-clustered immune cells. **c** Heatmap showing the top 5 differentially expressed genes for each of the 11 immune cell clusters. **d** Dendrogram highlighting the relationship among each of the immune cell sub-clusters. **e** Feature plots of selected genes to further classify immune cells as T cells (*Cd3e, Gata3*), NK cells (*Klrb1c*), dendritic cells (*Cd24a*), B cells (*Cd19*), macrophages (*Fcgr1*), and fibrocytes (*Col1a1*). **f** Feature plot of *C1qa*, which emerged as a differentially expressed gene between macrophage clusters I5 and I6. **g** Bar plots showing the distribution of cells from two independent experiments in each of the immune cell clusters. Symbols indicate individual cell distributions for each of the two experimental replicates (triangle, circle)

**Table 3 Summary of immune clusters and expressed genes**

| ID | Vehicle[a] | E2[a] | E2+PBDE[a] | Key genes | Cell types |
|---|---|---|---|---|---|
| I0 | 30.1 | 14.0 | 12.8 | Klrc1, Cd3e | NK cells/T cells |
| I1 | 29.3 | 2.3 | 2.6 | Cd3e, Cd3g, Cd3d, | T cells |
| I2 | 4.6 | 31.2 | 23.5 | H2-Ab1, Cd24a, Cd209a | Dendritic cells |
| I3 | 12.7 | 1.8 | 1.0 | Cd3e, Gata3 | T cells |
| I4 | 5.0 | 19.0 | 15.8 | H2-Ab1, Cd24a, Cd19 | Dendritic cells/B cells |
| I5 | 4.8 | 8.1 | 9.7 | H2-Ab1, Fcgr1, Adgre1 | Macrophage |
| I6 | 1.8 | 9.5 | 19.4 | H2-Ab1, Fcgr1, Adgre1, C1qa, C1qb, C1qc | Macrophage (M2) |
| I7 | 5.1 | 3.2 | 0.5 | Cd7e | T cells |
| I8 | 2.1 | 5.0 | 5.6 | Col1a1 | Fibrocytes |
| I9 | 2.0 | 2.7 | 7.1 | Cd3e | T cells |
| I10 | 2.7 | 3.2 | 2.0 | Col1a1 | Fibrocytes |

[a]Values are presented as percentages of total number of immune cells per treatment group (e.g., 30.1% of immune cells from Vehicle treated group are clustered in I0)

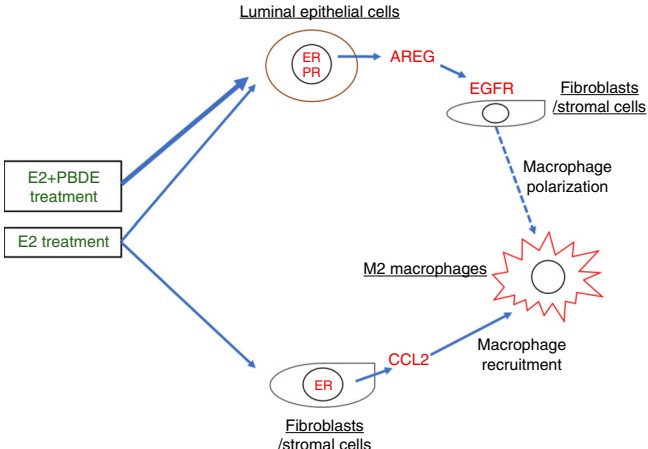

**Fig. 7** Proposed model summarizing two different mechanisms of M2 macrophage recruitment and polarization by E2 or E2 + PBDE treatment. E2 enriches ECM/fibroblast cells expressing *Ccl2*, which in turn recruits M2 macrophages. PBDE, in the presence of E2, increases *Areg* expression in *Esr1+Pgr+* luminal cells, which promotes M2 macrophage polarization through the E2–AREG–EGFR–M2 macrophage pathway

and TEB-like structures containing Ki67+ cells (Figs. 1c–e, 4e, f). By stratifying by lesion subtype, Ki67 was associated with higher risk among women with proliferative lesions with atypical hyperplasia[36].

Recently, several groups have used scRNAseq to decipher differentiation hierarchies and heterogeneity of mammary epithelial cells[12]. To the best of our knowledge, such studies have not assessed how hierarchy/heterogeneity is influenced by EDCs with hormones. Our scRNAseq data from 14,853 cells yielded 11 cell populations, including three major types (luminal epithelial, stromal/fibroblasts/ECM, and immune cells). Based on gene expression profiles, these clusters have been assigned to cell types found in the mammary gland (Fig. 2).

It has been a long-held belief that ER and PR are co-expressed in the same cells and PR is a marker of ER activation because PR is transcriptionally regulated by ER. However, ER and PR were reported to be expressed in different cells in human and mouse mammary glands[37]. According to our scRNAseq and IF analyses, mature (L0/L3) and intermediate luminal epithelial (L1) populations from mouse mammary glands comprise four different cell types: *Esr1+Pgr+* (DP), *Esr1+Pgr−*, *Esr1−Pgr+*, and *Esr1−Pgr−*

(DN). ScRNAseq allowed us to directly assess the gene profiles of these four cell types.

One scenario for the intermediate and mature luminal population is that E2 upregulates *Pgr* in both *Esr1+* and *Esr1−* cells through direct and paracrine manners. The administration of PBDEs together with E2 induced PR independently from ERα status (Table 2) leading to more cell differentiation in L1 and L0/L3, by promoting the expression of genes associated with pro-breast cancer pathways. Accordingly, the changes produced by E2 + PBDE on luminal cell populations at the single cell level correlated with our phenotypic observations on mammary gland whole mounts and IF analysis. Importantly, luminal progenitor cells (L2) had very low levels of *Esr1* and *Pgr*, implying their hormone-independent action.

E2 also increased the expression of *Areg* in luminal epithelial cells. AREG is a ligand of EGFR, whose transcript was present in the ECM/fibroblast clusters (Supplementary Fig. 12). AREG-EGFR cross-talk is necessary for ductal elongation and development[38]. Peterson et al. have suggested that AREG is a critical downstream effector of estrogen signaling in ER+ breast cancer[39]. In L0/L3 and L1, treatment with E2 and PBDEs increased the number of *Esr1+/Pgr+/Areg+* cells (Supplementary Fig. 11). These suggest that PBDEs, together with E2, facilitate ductal regrowth by influencing master regulators and potentially the ER+ breast cancer pathogenesis. Importantly, an *Esr1+* ECM/fibroblasts cluster (F2) appeared after E2, but not ER + PBDE treatment. The results suggest a selective effect of PBDEs on specific types of luminal epithelial cells together with E2.

Proper mammary gland development relies on precise coordination of bi-directional crosstalk between epithelial and immune cells. In the vehicle-treated glands, most immune cells were T and NK cells. These populations were reduced upon treatment with E2. To date, studies describing the function of T and NK cells in mammary development are lacking. In multiple sclerosis models, estrogen has been shown to inhibit CD4+ T cell expansion and increase T cell apoptosis[40]. Further studies will be necessary to determine the mechanisms linking E2-induced reduction of T and NK cell populations. Concomitantly, E2 shifted the immune cell population towards DCs and macrophages. Macrophages are key players in mammary gland elongation and branching[28]. The recruitment of M2 macrophages to the mammary duct is regulated by estrogen and progesterone[31]. M2-associated pro-tumoral functions include promotion of angiogenesis, matrix remodeling, and suppression of adaptive immunity[41]. M2 type macrophages were identified in our scRNAseq analysis as I6. Our results suggest that E2 + PBDE increased M2 macrophage populations; this activity might contribute to E2 + PBDE-induced luminal cell proliferation, and

possibly promote hormone receptor-positive breast cancer. Furthermore, E2 treatment increased B cell and DC populations (Table 3). It was recently reported that macrophages, DCs, and B lymphocytes increased in high mammographic density epithelium rather than low density tissue, linking to protumor inflammation[42].

Therefore, the results of our single-cell analysis indicate that E2 treatment induces redevelopment of luminal epithelial cells through the expression of AREG and EGFR+ fibroblasts, which mediate the recruitment of M2 macrophage (Fig. 7). Furthermore, E2 can enrich the $Esr1^+$ ECM/fibroblasts expressing $Ccl2$ to recruit M2 macrophages. Our results suggest that PBDEs promote the E2–AREG–EGFR–M2 macrophage pathway, but not the E2–CCL2–M2 macrophage pathway.

In this study, the PBDE-only samples were omitted because their mammary gland phenotype and RNAseq results were similar to those of vehicle. Although this could be one drawback of this study, technical limitation to isolate cells to accurately reflect small differences in mammary glands would prevent meaningful analysis of PBDE-only samples vs. vehicle samples.

In summary, treatment with external estrogen promotes mammary gland regrowth in a surgical menopause model. This regrowth increased TEB-like structures that contain Ki67 positive cells. PBDEs + E2 was found to increase the number of DP and $Esr1^-Pgr^+$ cells in two luminal epithelial populations (mature and intermediated luminal cells), with estrogen-regulated and cross-talk networks. Results also suggested that PBDE + E2 increased M2 macrophage populations, which could contribute to TEB-like structures increase and production of a pro-tumoral environment. Actively proliferating ducts have higher chances of DNA mutations; they are associated with increased breast cancer risk. Furthermore, co-administration of PBDEs with E2 increased $Esr1^+/Pgr^+/Areg^+$ cells in luminal epithelial population as well as $Egfr^+$ stromal/fibroblasts. Thus, PBDEs appear to facilitate the E2-mediated ductal regrowth activity and promotes breast cancer development through induction of key signal transduction pathways. It warrants further investigation to understand how progesterone treatment modulates the effects of E2 and PBDEs. Our results clarify how exposure to E2 and EDCs like PBDEs after menopause impact mammary glands. Ultimately, the findings advance understanding of how such exposure can increase the risk of developing breast cancer.

## Methods

**Chemicals**. BDE-47 [bromine substitution pattern 2,2′,4,4′], BDE-100 [2,2′,4,4′,6], and BDE-153 [2,2′,4,4′,5,5′] were purchased from AccuStandard, Inc. 17β-estradiol (E2) (Sigma-Aldrich Coporation, St. Louis, MO) and dimethyl sulfoxide (DMSO 99.7%; Sigma-Aldrich Corporation [St. Louis, MO]) were purchased.

**Animal**. Female BALB/cj mice were obtained from the Jackson Laboratory (Bar-Harbor, ME) and housed at the City of Hope Animal Resources Center. Mice were maintained on a 12 h light/dark cycle. All institutional guidelines for animal care and use were followed. All animal research procedures used in this study were approved by the Institutional Animal Care and Use Committee (IACUC) at City of Hope. Facilities are credited by Association for Assessment and Accreditation of Laboratory Animal Care (AAALAC), and operated according to NIH guidelines. Mice were housed in polypropylene cages. Water was filtered twice using reverse osmosis and carbon block system and was supplied ad libitum from glass bottles with a regular shack and nestlet. Supplied water was confirmed to have no detectable levels of estrogenic activity using AroER tri-screen system previously established in the lab[43]. We avoided using corn-cob bedding due to the report about estrogenic activity[44], and therefore mice were housed in only Sani-Chips beddings.

**Experimental design for PBDE exposure**. To examine the effects of E2 and PBDEs on mouse mammary gland structures, 9-week-old female BALB/cj mice were ovariectomized. Exposure was started 10 weeks (time to recover from surgery and to make sure estrogen levels have dropped sufficiently) after OVX (vehicle $n = 10$, E2 $n = 10$, PBDE $n = 10$, and E2 + PBDE $n = 10$). OVX mice were fed with diet supplemented with PBDE and/or E2 administration by injection for one week. E2

(1 μg/injection) was administered daily by interperitoneal injection. Our E2 treatment dosage produces stable levels of E2 (40–100 pg/mL)[45], which is equivalent to the serum levels of postmenopausal women[46]. To replicate physiologically relevant exposures, we assessed the relationship between dosage and both the ratio and murine sera levels of the three PBDEs congeners. The ratio was adjusted to reflect human exposure of three major PBDEs detected[47]. Compared to the highest concentration scenario in human exposure (i.e., maximum observed serum concentration), the mouse serum concentration in this study was 10–30 times higher than human exposure[47]. After one-week of treatment, mice were euthanized to collect blood and mammary glands.

**Mammary gland whole-mount analysis**. Glands were hydrated in decreasing concentrations of ethanol, and then stained in 0.025% Toluidine Blue. Afterwards, the slides were immersed in methanol, followed by ethanol, and a 4% ammonium molybdate solution. The glands were then dehydrated in increasing grades of ethanol, and cleared using Histoclear (National Diagnostics) overnight. Mounting was done using Permount (Fisher Chemical), and the glands were imaged using an EVOS FL Auto Imaging System (ThermoFisher Scientific). During examination, slides were de-identified, and TEB-like structures counted in a blinded manner.

**RNA sequencing and data analysis**. Total RNA was extracted from mammary glands using RNeasy Mini kit. Sequencing, processing, and sequence read mapping were conducted following the method used in our previous study[48]. All sequencing data were submitted to the GEO database. Ingenuity pathway analysis (IPA, Ingenuity® Systems, www.ingenuity.com) was used to identify the biological functions, pathways, and mechanisms for the filtered genes (fold change > 1.5, FDR < 0.05).

**Immunohistochemistry and mammary duct quantitative analysis**. Tissue sections were deparaffinized in xylene followed by decreasing grades of ethanol. Samples were then quenched in 3% hydrogen peroxide and pretreated to promote antigen retrieval by using a steaming method with citrate buffer solution (pH 6.0). Slides were incubated in Dako Protein Block (Agilent), and then incubated with primary antibody to KRT18 (Abcam, ab181597) at a dilution of 1:2000 or Ki67 (Cell Signaling Technology, 12202S) at a dilution of 1:200. KRT18 antibody was validated using mouse liver as positive control, and Ki67 antibody was validated using mouse intestine as positive control. The samples were washed and incubated in Dako Rabbit Polymer (Agilent). Slides were further incubated with chromogen diaminobenzidine tetrahydrochloride (DAB). For amplification and visualization of our KRT18 or Ki67 signal, slides were counterstained with hematoxylin, rehydrated, and then mounted. For duct quantification, slides were imaged on an EVOS FL Auto Imaging System (ThermoFisher Scientific) and analyzed using ImageJ (NIH). All KRT18 or Ki67 positively stained ducts were detected using the Color Threshold tool in ImageJ. Based on KRT18 IHC images, ducts were quantified and sorted into two categories, either large or small ducts as defined by a cutoff of 1906 μm². This cutoff was derived from taking the arithmetic mean of all ducts detected in the vehicle-treated group.

**IF analysis**. After the rehydration process, slides were immersed in water before being subjected to microwave treatment for antigen retrieval. For ER and PR staining, slides were placed in citrate buffer solution (pH 6.0) and microwaved. Slides were blocked with Antibody Diluent (Agilent, S302283-2). Samples were then incubated with primary antibodies. Tyramide signal amplificiation fluorophores (TSA) were used as the substrate to the HRP-conjugated antibody (Perkin Elmer, FP1497001KT, FP1487001KT). Slides were mounted using Vectashield Mounting Medium for Fluorescence with DAPI (Vector, H-1200). Primary antibodies: anti-ERα rabbit antibody diluted 1:500 (Millipore Sigma, 06-935); anti-PR rabbit antibody diluted 1:1000 (Abcam, ab131486); secondary antibodies: Mach 2 Rabbit HRP-polymer diluted 1:2 (Biocare Medical, RHRP520L) in Van Gogh Diluent (Biocare Medical, PD902L). ERα and PR antibodies were validated using B6C3F1 mouse uterus tissue (diestrus) by LaPlante et al.[49]. Images were captured on a Zeiss Observer II with a ×40/1.4NA Plan-Apochromat DIC Oil objective.

**Mammary gland dissociation into single-cell suspension**. We isolated cells from lymph node removed mammary glands in mice of three treatment groups (OVX, E2, and E2 + PBDE). Cells isolated from two biological replicates were analyzed separately. The mammary gland was cut into small strips of 5 mm thickness and digested with 1.5 mg/mL DNAse I (Millipore Sigma, #10104159001), 0.4 mg/mL Collagenase IV (Worthington, CLS-4, Lot: 47E17528A), 5% FBS, 10 mM HEPES in HBSS. The mixture was strained through a 70 μm cell strainer. 1 mL of ACK lysis buffer was used to remove residual red blood cells from the sample. Dead cells were removed using Dead Cells Removal Microbeads (Miltenyl Biotec).

**Single cell RNA-seq and analysis**. Cell number and viability were measured using a TC20 Automated Cell Counter (BioRad). We only processed samples showing at least 80% viability. Cells were then loaded onto the Chromium Controller (10x Genomics) targeting 2000–5000 cells per lane. The Chromium

v2 single cell 3′ RNA-seq reagent kit (10x Genomics) was used to process samples into single cell RNA-seq libraries according to the manufacturer's protocol. Libraries were sequenced with a Hiseq 2500 instrument (Illumina) with a depth of 50k–100k reads per cell. Raw sequencing data were processed using the 10x Genomics Cell Ranger pipeline (version 2.0) to generate FASTQ files and aligned to mm10 genome to gene expression count. The subsequent data analysis was performed using the Seurat package and R scripts. Cells with mitochondrial read rate > 10% and <200 detectable genes were considered as low-quality and filtered out. The samples from two batches were combined and analyzed using CCA implemented in Seurat package. Normalized and scaled data were clustered using the top principal components of HVGs. The t-SNE algorithm was used to visualize the resulting clusters. Cluster-specific markers were identified to generate heatmap and feature plots in the identified cell clusters. Gene set enrichment analysis (GSEA) was performed using genes ranked by the fold changes between different conditions to evaluate the significance activation of 50 HALLMARK gene sets in MSigDB[22] (http://software.broadinstitute.org/gsea/msigdb/collections.jsp). GSEA was also performed on the single cell level, using genes ranked by mean centered log2-normalized read counts and Hallmark gene sets in MSigDB. Enrichment scores of cells in each cluster were averaged and used to generate hierarchical clustering diagram using Cluster v and Java TreeView.

**Statistics and reproducibility**. For TEB quantification using whole-mounted mammary glands, data was presented as mean ± SEM, and Tukey's multiple comparisons test was performed. Statistics for KRT18 duct quantification was done using Mann–Whitney tests. For whole RNAseq statistics comparing expression levels of *Mki67*, *Pgr*, and *Areg*, Tukey's multiple comparisons test was used. ScRNAseq was done on duplicate single-cell preparations from two independent experiments.

**Reporting summary**. Further information on research design is available in the Nature Research Reporting Summary linked to this article.

## Data availability

The raw scRNAseq data generated in this study has been deposited in GEO with the accession code GSE125272. Top 20 differentially expressed genes for the overall clusters, luminal, stromal/fibroblast, and immune cells are provided as Supplementary Data 1–4, respectively. The source data underlying plots shown in Figs. 1 and 4 are shown in Supplementary Data 5.

## Code availability

Computational analyses were performed in R (version 3.5.1, available at https://www.r-project.org/). The software and analysis pipeline to pre-process raw data of The Chromium Single Cell Gene Expression are available at https://support.10xgenomics.com/single-cell-gene-expression/software/overview/welcome. Most analyses and visualization were performed using Seurat package (V3.0.1; available at https://satijalab.org/seurat/install.html) in support of GSVA (V1.30.0; https://bioconductor.org/packages/release/bioc/html/GSVA.html), ape (V5.3; http://ape-package.ird.fr/), ggplot2 (V3.2.0; https://ggplot2.tidyverse.org/) and cowplot (V0.9.4; https://wilkelab.org/cowplot/) packages. The scripts used for each analysis are available from the corresponding author upon reasonable request.

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

## Acknowledgements

As part of the Breast Cancer and Environmental Research Program, this work was supported by NIH U01ES026137-01 (MPIs Chen and Neuhausen) and a pilot grant from the City of Hope Center for Cancer and Aging. We thank the City of Hope Core Facilities, including the Integrative Genomics Core and Light Microscopy Digital Imaging Core, for the excellent technical support. The City of Hope Core Facilities are supported by the National Cancer Institute of the National Institutes of Health under award number P30CA033572. The authors would like to thank Ian Talisman, Ph.D. for editing the manuscript, Shawn Solomon and Peter Lee, M.D. for guidance on immunofluorescence analysis, and Dr. Jose Russo and his team for sharing expertize on mammary gland whole mounts staining and evaluation.

## Author contributions

N.K. and S.C. designed the study. N.K., G.C., L.B., H.-J.S., T.Y., and J.L. performed experiments. J.W. performed the 10X library production and sequencing. J.P. performed PBDE measurement. N.K., G.C, X.W., K.S., C.W., T.S., S.V., and S.C. analyzed data. N.K., G.C., X.W., K.S., and S.C. wrote the manuscript. M.R. and S.L.N. discussed the translation of the findings to the observations in humans. S.C. supervised research.

## Competing interests

The authors declare no competing interests.
