## [Peer Review File · Communications Biology]

Reviewers' comments:

Reviewer #1 (Remarks to the Author):

Kanaya et al perform scRNAseq to address the effects of an endocrine disrupting agent on the mammary gland at a stage that mimics menopause. While the data provide some new insights into hormone receptor positive cells in the mammary gland, the full implications of the data are difficult to interpret.

- It is not clear why PBDE was selected as the endocrine disruptor? Is there any evidence linking this to increased breast cancer risk?

- Can PBDE exposure affect mammary gland development during pubertal growth and post-puberty stages in the presence of physiological levels of estrogen? PBDE alone is not capable of causing ductal proliferation or the emergence of PR+ cells.

- PBDEs are known to cause oxidative stress-related damage including DNA damage, mitochondrial deregulation and apoptosis. Did authors observe cells undergoing these changes?

- The authors used an unbiased approach to interrogate total cells from the whole mammary using scRNAseq. However, in control samples, epithelial cells are under-represented. Please provide the number of cells in each cluster defined in t-SNE plots.

- Based on sc-RNAseq data, it is suggested that epithelial and macrophage populations undergo changes due to E2 and E2 + PBDE treatments. These changes need to be assessed and confirmed directly by enriching individual cell populations.

- Interestingly, partly based on bulk cell RNA-seq information, the authors decided not to test PBDE alone condition for sc-RNAseq. However, it should be noted that whole gland tissue including epithelial and non-epithelial cells were used to generate bulk cell DE gene profiles between different treatment conditions. This approach will generate information that is an average between the different cell lineages. Therefore, qPCR validation of DE genes between different cell lineages is required. Also, by not testing PBDE alone in a sensitive technique like sc-RNAseq, some crucial information may have been missed that can potentially provide molecular insight into how PBDE works.

- No insight is provided into the possible molecular mechanism by which PBDE causes a synergistic effect with E2 treatment.

- E2 + PBDE treatment causes proliferation, the generation of small TEB-like structures and an increase in PR+ cells. In Figure 3C, it appears that a ductal section from E2 is compared with a TEB-like structure in the E2+PBDE condition which could potentially bias the quantitative assessment of PR+ cells between the two conditions. It would be helpful to quantify the change in Ki67+ cells either by FACS or some other method.

- Interestingly, partly based on bulk cell RNA-seq information, authors decided not to test PBDE alone condition for sc-RNAseq. However, it should be noted that whole gland tissue including epithelial and non-epithelial cells were used to generate bulk cell DE gene profiles between different treatment conditions. This approach, which authors also agree, will generate information that will be an average between different cell lineages. Therefore, qPCR validation of DE genes between different cell lineages is required. Also, by not testing PBDE alone condition by a sensitive technique like sc-RNAseq, authors might have missed some crucial information that can potentially provide molecular insight into how

PBDE works?

- Figure 2C - the dendrogram to show hierarchical clustering between cell populations is missing in the heatmap.

- the results are very descriptive with lists of genes presented in some sections.

Reviewer #2 (Remarks to the Author):

In this study the authors investigate the impact of 17β -estradiol (E2) and polybrominated diphenyl ethers (PBDEs) on mammary epithelial cells using scRNASeq. Although the study is conceptually interesting there are major issues that I think the authors need to address before publication:

PBDE effects on mouse mammary glands:

1) On P7 TEB-like structures. In my understanding TEBs only exist during puberty and yes they refer to them as TEB-like but I am still not sure if that would be correct. Also they refer to F1b for this and I find it hard to spot anything there, might be useful if the authors highlighted the structures and quantify them.

2) "Moreover, E2-present groups contained a significant number of TEB-like structures and more ducts; the ducts were often associated with the TEB-like structures"
Judging number of ducts is really hard from the images. The vehicle and PBDE also appear to have ducts but they just seem a bit thinner/fainter? Again quantification is needed.

3) "Importantly, mammary fat pads from the E2-present groups exhibited a marker of cellular proliferation, Ki67, in the TEB-like structures (Fig. 1c)." The authors should present Ki-67 stainings for the other conditions as well.

Transcriptome analysis using whole mouse mammary glands

4) The authors should mention at which threshold (e.g. FDR <0.01) DE genes were called. In general this section doesn't add much to the entire manuscript.

5) "cytokine profiling was performed using the differentially expressed gene" this is a bit confusing as it suggests that cytokines were actually measured which was not the case.

6) "Compared to vehicle, Il10 was increased 1) in the E2 group with a fold change of 2.07; and 2) in the E2 + PBDE group with a fold change of 2.35. IL10 has been identified as M2 macrophage polarization stimuli¹⁵. Thus, we hypothesize that addition of PBDEs to E2 may increase the number of M2 macrophages and/or their activation in mammary glands." If the hypothesis that there is more IL10 in E2+PBDE in comparison to E2 alone then the logFC plus p-value from this comparison should be mentioned at this point. It is impossible to know for the reader whether LFC from 2.07 to LFC from 2.35 is significant or not.

Single-cell RNA sequencing approach to determine the effects of E2 and PBDEs on mouse mammary glands

7) The scRNAseq analysis has several flaws that need to be corrected before the manuscript is published.

8) "Previous scRNAseq studies of the mammary gland focused individually on either epithelial cells or immune cells^{7,8}. In contrast, we have developed a protocol that allows us to isolate and assess both cell populations simultaneously." Developed a protocol is a bit of an overstatement given that the authors simply omitted FACS before performing 10x.

9) Given the data was produced in two batches the authors should include a plot that shows that the CCA batch correction produced sensible results, e.g. a tSNE before and after correction.

10) In general, the description of cluster is a bit too in depth for the flow of the paper. It might be beneficial to only keep clusters that are relevant for the story in the main text and move the rest to the supplemental material.

11) "C2 could be related to C0 as both clusters have many *Esr1*-positive cells, and are adjacent to each other on the tSNE plot." Adjacency of clusters in a tSNE plot does not imply similarity. C0 is also close to C6 and C7 in the tSNE plot which represent completely different celltypes. This an over-interpretation

12) "Based these data and analyses, we concluded that C4 contained mature luminal epithelial cells whereas C5 contained both mature and luminal progenitor cells." This statement is illogical assuming the authors have performed clustering that resolves all cell types into individual clusters, otherwise the C5 couldn't contain mature and progenitor cells. Further, the author shows in Figure 2c that it is in fact the same cells that express mature and progenitor markers, suggesting instead that these are either committed progenitors or intermediate cells. This needs reanalysis.

13) "Furthermore, C4 and C5 were thought to contain cells expressing genes promoting cancer development, such as those found in the PI3K_AKT_MTOR_SIGNALING, MTORC1_SIGNALING, REACTIVE_OXIGEN_SPECIES_PATHWAY, and MYC_TARGETS gene sets." Not sure what the point of this statement is. Don't all cells express some genes that are involved in cancer? Tp53, Ras, Braf, mTOR?

14) The way the impact of compound treatment was assessed analytically is insufficient. Table2 is (presumably) displaying mean cluster frequencies per condition, the author should include SD and consider assessing the statistical significance of any of the claimed changes.

15) Double positive (*Esr1*+/*Pgr*+) and *Pgr*+ cells increased after administration of E2 + PBDE

The computational analysis in this section is insufficient. First, the authors speak of cells being "positive" for certain receptors, this concept is well defined on the protein level, specifically for any methods involving antibody-based detection. For scRNAseq this is not the case, absence of a transcript does not imply "negativity" on the protein level and presence does not imply "positivity". Instead, it would be more sensible to discuss what is actually compared, mean expression levels per cluster. Further, none of the results are accompanied by any effect size (logFC) or statistical significance, making it impossible for the reader to interpret the data. The authors should simply perform a differential expression analysis between their conditions of interest instead of picking certain genes to look at.

16) "ER α and/or PR localized in the nucleus (Fig. 3c). Another effect of the treatments was on the proliferation marker Ki67. While E2 increased the number of Ki67-positive cells over vehicle, the

addition of PBDEs to E2 resulted in significantly more both PR-positive and Ki67-positive cells (Fig. 3c, right). These results are consistent with our single cell data for C4 and C5." It is impossible for the reader to judge the significance of the claims based on the provided images especially without quantification.

17) "Results from our scRNA profiling analysis (Fig. 3d) and our IF staining (Fig. 3c) strongly indicate that these highly proliferative cells in C4 and C5 were probably present in TEB-like structures associated with mammary gland regrowth following" The association of cells from scRNAseq to a region in the gland, I.e. "TEB-like structures", is a big stretch. please reword.

18) Functional characterization of Esr1- and Pgr-expressing cells

Dividing clusters into subgroups of cells based on the expression of two genes (Pr, Esr1) provides as expected no extra information. If DP, DN Esr1+ and Pr+ cells were distinct groups surely they would cluster separately in their analysis. As they don't, it is not surprising that the treatment effects are almost identical between all of the defined subgroups within one cluster (F3e left panel).

19) "C5: DN cells from all three treatments were less transcriptionally active than cells expressing Esr1 and/or Pgr. " Less transcriptionally active? Do they produce less RNA?

20) In addition, the authors should check whether the DN cells have a smaller library size and are therefore classified as "DN" and have seemingly lower expression of the analysed pathways.

21) For C5 as the authors suggest the hormone receptor status appears to explain more variability than the treatment. Eyeballing the tSNE plot it appears that there might be two groups of cells (eg. HR+, HR-) in C5, it might be sensible to check whether C5 can be subdivided by clustering (and not expression of 3 genes). The subdivision of C5 into progenitor, HR+ progenitors would reflect the clusters that previously observed in other scRNAseq papers (Pal et al., Bach et al.). After the clusters have been properly resolved the authors should again analyse the effect of treatment.

22) The section about upregulation of Areg is unnecessarily convoluted. If the hypothesis is E2 upregulates Areg, why not test for DE and see if Areg is upregulated? The entire section would benefit from a DE test between conditions.

23)scRNA sequencing identified 7 clusters of immune cells

This section is interesting but as above the authors should perform statistical test, I.e. differential expression analysis or differential abundance to show treatment effects.

24) Discussion

"Importantly, the scRNA profiling analysis identified a group of cells in C4 and C5 to be associated with TEB- like structures shown by IF imaging, i.e., cells were highly proliferative, positive for Mki67 and Top2a, and negative for Esr1 (Fig. 3d)." As stated above, a single IF image of Ki67 and a tSNE with the same marker is hardly enough to definitely relate cells to a particular location in the gland. Therefore, this is an over statement.

25) "To the best of our knowledge, we are the first to provide important evidence that functional differences exist between these cell types." No functional evidence was provided.

26) "MSigDB analysis revealed that DP cells were more functionally active compared to other cell types." What does functionally active mean?

27) "Our scRNA analysis revealed that E2 induced and PBDEs enhanced the expression of Gata3, mainly in C4 and C5 (higher in C4 which has more mature luminal cells). Gata3+ cells were also found in C7 and C8." Where exactly was this revealed?

28) Figure 5: This figure needs clearer labeling.

Reviewers' comments:

Reviewer #1:

Kanaya et al perform scRNAseq to address the effects of an endocrine disrupting agent on the mammary gland at a stage that mimics menopause. While the data provide some new insights into hormone receptor positive cells in the mammary gland, the full implications of the data are difficult to interpret.

Response: We have revised the manuscript significantly to clearly point out the significance of this study.

Comment 1: It is not clear why PBDE was selected as the endocrine disruptor? Is there any evidence linking this to increased breast cancer risk?

Response to comment 1: Exposure to higher levels of PBDEs, individually or in combination, has been linked to increased breast cancer risk. We added one sentence and a new reference in the Introduction to further elaborate on this point (Lines: 62-64).

Comment 2: Can PBDE exposure affect mammary gland development during pubertal growth and post-puberty stages in the presence of physiological levels of estrogen? PBDE alone is not capable of causing ductal proliferation or the emergence of PR+ cells.

Response to comment 2: Although the focus of our study was not PBDE exposure during puberty or post-puberty, reported epidemiological studies showed PBDE levels associated with the timing of puberty onset or menarche. There were no papers to show that PBDE exposure affects mammary gland development during pubertal growth and post-puberty stages in the presence of physiological levels of estrogen.

Comment 3: PBDEs are known to cause oxidative stress-related damage including DNA damage, mitochondrial deregulation and apoptosis. Did authors observe cells undergoing these changes?

Response to comment 3: Although the developmental neurotoxic effects of PBDEs have been reported (i.e., Costa et al., Toxicol Lett 2014, 230: 282-94), we did not observe similar changes in mammary glands during menopausal transition or menopause. Our single-cell analysis has found that the major targets of PBDEs are luminal epithelial cells, probably acting through ER α .

Comment 4: The authors used an unbiased approach to interrogate total cells from the whole mammary using scRNAseq. However, in control samples, epithelial cells are under-represented. Please provide the number of cells in each cluster defined in t-SNE plots.

Response to comment 4: The number of cells in each cluster defined in the t-SNE plots are provided in Supplementary Table 2. To evaluate epithelial cells and mammary ductal structures in the mammary glands in their entirety, we quantified the ducts after KRT18 immunohistochemistry staining of epithelial cells. Ducts were classified as either small or large

ducts (Fig. 1d). Although total numbers of ducts were similar for all four groups, the E2 and E2 + PBDE groups had more developed (large) ducts. We also provided representative mammary gland images (Supplementary Fig. 1) for readers to review. We further assessed the general impact of vehicle, E2, or E2 + PBDE on cluster distribution (Figs. 2e and 2f, Table 1).

We agree with this reviewer that fewer epithelial cells in the vehicle samples were analyzed. This under-representation could result from technical limitations to isolate epithelial cells from small/regressed ducts in ovariectomized mice, which also had an absence of TEB-like structures in their mammary glands. We have re-clustered luminal epithelial cells – the main target cells of E2 and PBDEs. Importantly, although the vehicle group had fewer luminal cells, the distribution of the cells in different luminal clusters was similar to those after E2 or E2+PBDE treatments (Supplementary Fig. 8a), which meant that at least some parts of the epithelial ductal structure were digested and captured. We also provided the number of luminal epithelial cells by treatment and by hormone receptor status in Table 2. Accordingly, the text has been modified to elaborate on these points (Lines: 234-252).

Comment 5: Based on sc-RNAseq data, it is suggested that epithelial and macrophage populations undergo changes due to E2 and E2 + PBDE treatments. These changes need to be assessed and confirmed directly by enriching individual cell populations.

Response to comment 5: We re-clustered luminal epithelial and immune cells and compared their distribution by treatments (Lines: 234-252, 319-348). The re-clustering of luminal epithelial cells confirmed that E2 and E2 + PBDE treatments increased all types of luminal cells (Lines: 248-252 and Fig. 3a-h) (Supplementary Table 3). Immunohistochemistry analyses of KRT18 (Fig. 1d) and immunofluorescence analysis of HRs (Fig. 4d) further supported the changes in luminal epithelial cells revealed by scRNAseq analysis. Re-clustering of immune cells confirmed that E2 + PBDE treatment increased M2 macrophage cluster (I6), and, moreover, I6 had a higher score for the IL10 pathway (Lines: 342-348, Fig. 6a-g, Supplementary Fig. 9c and 13). The upregulation of IL10 following E2 + PBDE treatment, was also supported by our mammary gland bulk-RNAseq analysis.

Comment 6: Interestingly, partly based on bulk cell RNA-seq information, the authors decided not to test PBDE alone condition for sc-RNAseq. However, it should be noted that whole gland tissue including epithelial and non-epithelial cells were used to generate bulk cell DE gene profiles between different treatment conditions. This approach will generate information that is an average between the different cell lineages. Therefore, qPCR validation of DE genes between different cell lineages is required. Also, by not testing PBDE alone in a sensitive technique like sc-RNAseq, some crucial information may have been missed that can potentially provide molecular

Response to comment 6: We understand and appreciate these comments. We did not include PBDE-only conditions for our scRNAseq analysis since there was a lack of phenotypic changes in the PBDE-only group versus the vehicle treated group when evaluated by mammary gland whole mount analysis, and limited gene expression changes from whole gland RNAseq analysis when compared to the vehicle samples. For the revision, we have quantified mammary ducts using KRT18 IHC staining (Fig. 1d) and confirmed the lack of differences between the PBDE-

only samples and the vehicle samples. We agree with reviewer's comments that scRNAseq can be a sensitive technique to detect gene expression changes caused by PBDE treatment alone, but we were unsure whether current technical constraints would prevent meaningful analysis of the limited differences between PBDE-only samples and vehicle samples. We elaborated on our rationale for the exclusion of PBDE only samples in the discussion section, and we do recognize that as a potential limitation of this study (Lines: 428-432).

Comment 7: No insight is provided into the possible molecular mechanism by which PBDE causes a synergistic effect with E2 treatment.

Response to comment 7: When we compared E2 to E2+PBDE treated samples, the main differences were increases in luminal epithelial cells and in the M2 macrophage cluster. Based on our new analyses of our scRNAseq data, we proposed that E2 treatment induces the redevelopment of luminal epithelial cells and ductal development potentially through the expression of AREG and the establishment of EGFR+ fibroblasts that mediate the recruitment of M2 macrophages, via the E2-AREG-EGFR-M2 macrophage pathway (Fig. 7), which seemed to be further augmented by the addition of PBDE. On the other hand, E2 can also enrich the *Esr1*+ ECM/fibroblasts expressing *Ccl2* to recruit M2 macrophages. This was not observed in the E2+PBDE group. Therefore, our results suggest that PBDEs promote the E2-AREG-EGFR-M2 macrophage pathway, but not the E2-CCL2-M2 macrophage pathway (Lines: 422-427).

Comment 8: E2 + PBDE treatment causes proliferation, the generation of small TEB-like structures and an increase in PR+ cells. In Figure 3C, it appears that a ductal section from E2 is compared with a TEB-like structure in the E2+PBDE condition which could potentially bias the quantitative assessment of PR+ cells between the two conditions. It would be helpful to quantify the change in Ki67+ cells either by FACS or some other method.

Response to comment 8: We agree with your comment. We did analyze Ki67+ ducts by IHC staining of mammary glands and found an increase of Ki67+ ducts and TEB-like structures after E2 or E2 + PBDE treatment. We included the results as Fig. 4e and revised the text (Lines: 270-271).

Comment 9: Figure 2C - the dendrogram to show hierarchical clustering between cell populations is missing in the heatmap.

Response to comment 9: Thank you so much for pointing this out. We updated Fig. 2 (Fig. 2c) to add hierarchical clustering and added dendrograms to other figures as well (Fig. 3d, 5d, and 6d). We believe that these changes clarified the relationships between clusters, which would help the readers' understanding.

Comment 10: the results are very descriptive with lists of genes presented in some sections.

Response to comment 10: We updated the entire manuscript, especially in the section regarding changes to the mammary gland structure at the single cell resolution, to focus on the interpretation of the results associated with luminal epithelial cells, fibroblasts/stromal cells, and

immune cells. We added more discussion on previous findings that support the E2-AREG-EGFR-M2 macrophage pathway and the E2-CCL2-M2 macrophage pathway.

Reviewer #2:

In this study the authors investigate the impact of 17 β -estradiol (E2) and polybrominated diphenyl ethers (PBDEs) on mammary epithelial cells using scRNASeq. Although the study is conceptually interesting there are major issues that I think the authors need to address before publication:

We want to thank this reviewer for the valuable comments. We have extensively revised the manuscript to address these comments.

PBDE effects on mouse mammary glands:

Comment 1: On P7 TEB-like structures. In my understanding TEBs only exist during puberty and yes they refer to them as TEB-like but I am still not sure if that would be correct. Also they refer to F1b for this and I find it hard to spot anything there, might be useful if the authors highlighted the structures and quantify them.

Response to comment 1: We quantified the TEB-like and duct structures by three independent investigators. There were significant differences between the E2 and E2+PBDE groups for TEB counts (Fig. 1c) and the number of large ducts (Fig. 1d). The text was also updated accordingly (Lines: 112-115).

Comment 2: “Moreover, E2-present groups contained a significant number of TEB-like structures and more ducts; the ducts were often associated with the TEB-like structures” Judging number of ducts is really hard from the images. The vehicle and PBDE also appear to have ducts but they just seem a bit thinner/fainter? Again quantification is needed.

Response to comment 2: We quantified the number of TEB-like structures. There was a significant difference between the E2 and E2+PBDE groups for TEB counts (Fig. 1c). The text was updated accordingly (Lines: 100-103). To evaluate epithelial cells and mammary ductal structures in the mammary glands in their entirety, we quantified the ducts after KRT18 immunohistochemistry staining of epithelial cells. Ducts were classified as either small or large ducts (Fig. 1d). Although the total numbers of ducts regardless of size were similar among all four groups, the E2 and E2 + PBDE groups had more developed (large) ducts. We also provided mammary gland images (Supplementary Fig. 1) for readers to review.

Comment 3: “Importantly, mammary fat pads from the E2-present groups exhibited a marker of cellular proliferation, Ki67, in the TEB-like structures (Fig. 1c).” The authors should present Ki-67 staining for the other conditions as well.

Response to comment 3:

We have analyzed Ki67+ ducts by IHC staining of mammary glands and found an increase of Ki67+ ducts and TEB-like structures after E2 or E2 + PBDE treatment. The results are shown in Fig. 4e of this revised manuscript.

Transcriptome analysis using whole mouse mammary glands

Comment 4: The authors should mention at which threshold (e.g. FDR <0.01) DE genes were called. In general, this section doesn't add much to the entire manuscript.

Response to comment 4: Thank you for the suggestion. Genes differentially expressed between conditions were identified by a fold change > 1.5 and a P value <=0.05 (Lines: 121-122).

Comment 5: “cytokine profiling was performed using the differentially expressed gene” this is a bit confusing as it suggests that cytokines were actually measured which was not the case.

Response to comment 5: We updated the text. The current sentence reads: “...cytokine gene expression was compared among groups using the differentially expressed gene sets” (Lines: 142-143).

Comment 6: “Compared to vehicle, Il10 was increased 1) in the E2 group with a fold change of 2.07; and 2) in the E2 + PBDE group with a fold change of 2.35. IL10 has been identified as M2 macrophage polarization stimuli¹⁵. Thus, we hypothesize that addition of PBDEs to E2 may increase the number of M2 macrophages and/or their activation in mammary glands.” If the hypothesis that there is more Il10 in E2+PBDE in comparison to E2 alone then the logFC plus p-value from this comparison should be mentioned at this point. It is impossible to know for the reader whether LFC from 2.07 to LFC from 2.35 if significant or not.

Response to comment 6: We added the *p*-values (Lines: 146-147).

Single-cell RNA sequencing approach to determine the effects of E2 and PBDEs on mouse mammary glands

The scRNAseq analysis has several flaws that need to be corrected before the manuscript is published.

Comment 7: “Previous scRNAseq studies of the mammary gland focused individually on either epithelial cells or immune cells^{7,8}. In contrast, we have developed a protocol that allows us to isolate and assess both cell populations simultaneously.” Developed a protocol is a bit of an overstatement given that the authors simply omitted FACS before performing 10x.

Response to comment 7: Sentences were changed to “Previous scRNAseq studies on mammary glands focused individually on epithelial or immune cells^{8,9}. In this study, we have tried to assess both cell populations along with surrounding stromal cells altogether.” (Lines; 159-161).

Comment 8: Given the data was produced in two batches the authors should include a plot that

shows that the CCA batch correction produced sensible results, e.g. a tSNE before and after correction.

Response to comment 8: Thank you for your comment. In Supplementary Fig. 7, we have shown results of CCA before and after subspace alignment. As we observed, differences between batches were effectively removed in ACC (aligned canonical correlation) spaces compared to unaligned CC spaces. In the tSNE plot, we visualized that two batches were evenly distributed after clustering using ACC dimensions. Moreover, for readers to realize that the difference between batches, or biological replicates, did not compromise our conclusion, we put error bars on graphs showing cluster distribution (Figs. 2f, 3h, 5c and 6g, and Supplementary Fig. 10). We also included Supplementary Figs. 7f and 9 to show cluster distributions in each batch.

Comment 9: In general, the description of cluster is a bit too in depth for the flow of the paper. It might be beneficial to only keep clusters that are relevant for the story in the main text and move the rest to the supplemental material.

Response to comment 9: We have rearranged and revised the text significantly. We have also performed additional re-clustering analyses on luminal epithelial cells, fibroblasts, and immune cells and present the results separately. With thoughtful comments by this reviewer, this should be a significantly improved manuscript.

Comment 10: “C2 could be related to C0 as both clusters have many Esr1-positive cells, and are adjacent to each other on the tSNE plot.” Adjacency of clusters in a tSNE plot does not imply similarity. C0 is also close to C6 and C7 in the tSNE plot which represent completely different celltypes. This an over-interpretation.

Response to comment 10: Thanks. We have deleted this sentence.

Comment 11: "Based these data and analyses, we concluded that C4 contained mature luminal epithelial cells whereas C5 contained both mature and luminal progenitor cells." This statement is illogical assuming the authors have performed clustering that resolves all cell types into individual clusters, otherwise the C5 couldn't contain mature and progenitor cells. Further, the author shows in Figure 2c that it is in fact the same cells that express mature and progenitor markers, suggesting instead that these are either committed progenitors or intermediate cells. This needs reanalysis.

Response to comment 11: We have re-clustered C4 and C5 and into L0-L3 (L0 and L3; mature, L1; intermediate and L2; progenitor). We added a new section for these new analyses (Results, Re-clustering of luminal cells and their changes by treatment, Lines 234-252).

Comment 12: “Furthermore, C4 and C5 were thought to contain cells expressing genes promoting cancer development, such as those found in the PI3K_AKT_MTOR_SIGNALING, MTORC1_SIGNALING, REACTIVE_OXIGEN_SPECIES_PATHWAY, and MYC_TARGETS gene sets.” Not sure what the point of this statement is. Don't all cells express some genes that are involved in cancer? Tp53, Ras, Braf, mTOR?

Response to comment 12: Thanks. We deleted the sentences.

Comment 13: The way the impact of compound treatment was assessed analytically is insufficient. Table 2 is (presumably) displaying mean cluster frequencies per condition, the author should include SD and consider assessing the statistical significance of any of the claimed changes.

Response to comment 13: Thank you for your valuable comment. After effective removal of batch effects, two biological replicates showed similar changes after treatment as we claimed. We feel that performing statistical analyses on these two biological replicates is technically possible but less scientifically significant. Therefore, we decided to show consistency between two replicates by adding cluster distribution with error bars and cluster distribution in each cluster as explained in “Response to comment 8” above.

Comment 14: Double positive (*Esr1*+/*Pgr*+) and *Pgr*+ cells increased after administration of E2 + PBDE

The computational analysis in this section is insufficient. First, the authors speak of cells being “positive” for certain receptors, this concept is well defined on the protein level, specifically for any methods involving antibody-based detection. For scRNAseq this is not the case, absence of a transcript does not imply “negativity” on the protein level and presence does not imply “positivity”. Instead, it would be more sensible to discuss what is actually compared, mean expression levels per cluster. Further, none of the results are accompanied by any effect size (logFC) or statistical significance, making it impossible for the reader to interpret the data. The authors should simply perform a differential expression analysis between their conditions of interest instead of picking certain genes to look at.

Response to comment 14: We agree with this comment that *Esr1*+ cells are not necessary positive for an ER α receptor. We tried to demonstrate the PR receptor expression in ER α + and ER α - cells by immunofluorescence analysis (Fig. 4d). The expression of *Pgr* and *Areg* is upregulated by activated ER and we detected *Pgr*+ and *Areg*+ cells in mainly the E2 and E2 + PBDE treated samples (Fig. 4a and 4b). These results suggest that *Esr1*+ cells have functional ER. To address the reviewer’s comments, we have performed additional analyses on luminal epithelial cells (Lines 234-294).

Differential expression analysis is a possible alternative to address the difference between treatments. However, we realized DE analysis by treatments, including multiple clusters, ended up as a simple reflection of the difference in cluster distribution between treatments. On the other hand, DE analysis by treatments in single cluster did not lead to sensible results in most cases since cells in one cluster are similar in nature as this reviewer pointed out in Comment 17. This is why we put more focus on the differences in cluster distribution, which mainly indicates the difference in differentiation status or HR expression in mammary luminal cells.

Comment 15: “ER α and/or PR localized in the nucleus (Fig. 3c). Another effect of the treatments was on the proliferation marker Ki67. While E2 increased the number of Ki67-positive cells over vehicle, the addition of PBDEs to E2 resulted in significantly more both PR-

positive and Ki67-positive cells (Fig. 3c, right). These results are consistent with our single cell data for C4 and C5.” It is impossible for the reader to judge the significance of the claims based on the provided images especially without quantification.

Response to comment 15: The discussion on Ki67+ cells has been updated. Quantification was done using IHC slides stained for Ki67 in which we found an increase in Ki67+ ducts in the E2 and E2 + PBDE treated mammary glands compared to vehicle (Fig. 4e).

Comment 16: “Results from our scRNA profiling analysis (Fig. 3d) and our IF staining (Fig. 3c) strongly indicate that these highly proliferative cells in C4 and C5 were probably present in TEB-like structures associated with mammary gland regrowth following” The association of cells from scRNAseq to a region in the gland, I.e. “TEB-like structures”, is a big stretch. please reword.

Response to comment 16: We deleted this statement.

Comment 17: Functional characterization of Esr1- and Pgr-expressing cells

Dividing clusters into subgroups of cells based on the expression of two receptor genes (*Pgr*, *Esr1*) provides as expected no extra information. If DP, DN Esr1+ and Pr+ cells were distinct groups surely they would cluster separately in their analysis. As they don't, it is not surprising that the treatment effects are almost identical between all of the defined subgroups within one cluster (F3e left panel).

Response to comment 17: This discussion has been removed. We have re-analyzed luminal epithelial cells into four clusters (L0 to L3). L2 contains luminal progenitor cells that are negative for *Esr1* and *Pgr* expression (Fig. 3e). Luminal intermediate cells (L1) have more expressed transcripts (Fig. 3c and 3f). A new section has been added to discuss findings from re-clustering of luminal epithelial cells.

Comment 18: “C5: DN cells from all three treatments were less transcriptionally active than cells expressing Esr1 and/or Pgr. ” Less transcriptionally active? Do they produce less RNA?

Response to comment 18: This section has been rewritten with new analysis of luminal epithelial cells. According to comment 17 by this reviewer, comparison between DP, DN *Esr1*+ and *Pgr*+ in one cluster was removed.

Comment 19: In addition, the authors should check whether the DN cells have a smaller library size and are therefore classified as “DN” and have seemingly lower expression of the analysed pathways.

Response to comment 19: Similar to comment 18, we removed description about the comparison between DP, DN *Esr1*+, and *Pgr*+ cells.

Comment 20: For C5 as the authors suggest the hormone receptor status appears to explain more variability than the treatment. Eyeballing the tSNE plot it appears that there might be two

groups of cells (eg. HR+, HR-) in C5, it might be sensible to check whether C5 can be subdivided by clustering (and not expression of 3 genes). The subdivision of C5 into progenitor, HR+ progenitors would reflect the clusters that previously observed in other scRNAseq papers (Pal et al., Bach et al.). After the clusters have been properly resolved the authors should again analyze the effect of treatment.

Response to comment 20: We have performed new clustering of luminal epithelial cells, carried out careful analyses (Fig. 3 and 4), and discussed the findings on luminal progenitor, intermediate, and mature cells. We really appreciate your comments.

Comment 21: The section about upregulation of Areg is unnecessarily convoluted. If the hypothesis is E2 upregulates Areg, why not test for DE and see if Areg is upregulated? The entire section would benefit from a DE test between conditions.

Response to comment 21: AREG plays critical roles in mammary gland developed and its expression has been reported by a number of investigators to be upregulated by estrogen. Through consideration of this comment, we have revised the discussion on *Areg* significantly and now present an E2-AREG-EGFR-M2 macrophage regulatory pathway based on our results and those published by others (Lines: 422-427).

To further support our scRNAseq results, we have observed an increase of *Areg* expression in RNAseq analysis of whole mammary glands after E2 and E2 + PBDE treatments compared to the vehicle samples (Supplementary Fig. 2).

Comment 22: scRNA sequencing identified 7 clusters of immune cells

This section is interesting but as above the authors should perform statistical test, i.e. differential expression analysis or differential abundance to show treatment effects.

Response to comment 22: This section has been rewritten, and graphs showing cluster distribution were added (Fig. 6g and Supplementary Fig. 9c). We did not perform DE analysis by treatment as stated in our response to comment 14.

Comment 23: Discussion

“Importantly, the scRNA profiling analysis identified a group of cells in C4 and C5 to be associated with TEB- like structures shown by IF imaging, i.e., cells were highly proliferative, positive for Mki67 and Top2a, and negative for Esr1 (Fig. 3d).” As stated above, a single IF image of Ki67 and a tSNE with the same marker is hardly enough to definitely relate cells to a particular location in the gland. Therefore, this is an over statement.

Response to comment 23: Through consideration of this comment, the above statement has been removed.

Comment 24: "To the best of our knowledge, we are the first to provide important evidence that functional differences exist between these cell types." No functional evidence was provided.

Response to comment 24: This statement has been removed.

Comment 25: “MSigDB analysis revealed that DP cells were more functionally active compared to other cell types.” What does functionally active mean?

Response to comment 25: This statement has been removed after extensive revision of the manuscript.

Comment 26: “Our scRNA analysis revealed that E2 induced and PBDEs enhanced the expression of *Gata3*, mainly in C4 and C5 (higher in C4 which has more mature luminal cells). *Gata3*⁺ cells were also found in C7 and C8.” Where exactly was this revealed?

Response to comment 26: Luminal epithelial cells have been re-clustered and analyzed. The statement on *Gata3* has been removed.

Comment 27: Figure 5: This figure needs clearer labeling.

Response to comment 27: This figure has been removed. Our hypothesis is now shown in Fig. 7. We proposed that E2 treatment induces the redevelopment of luminal epithelial cells and ductal development possibly through the expression of AREG and establishment of EGFR⁺ fibroblasts that mediate the recruitment of M2 macrophage, i.e., the E2-AREG-EGFR-M2 macrophage pathway (Fig. 7), which seemed to be further augmented by addition of PBDE. On the other hand, E2 can also enrich the *Esr1*⁺ ECM/fibroblasts expressing *Ccl2* to recruit M2 macrophages. This was not observed in E2+PBDE group. Conclusively, our results suggest that PBDEs promote the E2-AREG-EGFR-M2 macrophage pathway, but not the E2-CCL2-M2 macrophage pathway. (Lines: 422-427).

REVIEWERS' COMMENTS:

Reviewer #1 (Remarks to the Author):

Most of my comments have been addressed satisfactorily.